# IL-1-dependent enteric gliosis guides intestinal inflammation and dysmotility and modulates macrophage function

Reiner Schneider [1], Patrick Leven[1], Shilpashree Mallesh[1], Mona Breßer[1], Linda Schneider[1], Elvio Mazzotta[2], Paola Fadda [2], Tim Glowka [1], Tim O. Vilz[1], Philipp Lingohr[1], Jörg C. Kalff[1], Fievos L. Christofi[2,3] & Sven Wehner [1,3✉]

*Muscularis Externa* Macrophages (*ME*-Macs) and enteric glial cells (EGCs) are closely associated cell types in the bowel wall, and important interactions are thought to occur between them during intestinal inflammation. They are involved in developing postoperative ileus (POI), an acute, surgery-induced inflammatory disorder triggered by IL-1 receptor type I (IL1R1)-signaling. In this study, we demonstrate that IL1R1-signaling in murine and human EGCs induces a reactive state, named enteric gliosis, characterized by a strong induction of distinct chemokines, cytokines, and the colony-stimulating factors 1 and 3. Ribosomal tagging revealed enteric gliosis as an early part of POI pathogenesis, and mice with an EGC-restricted IL1R1-deficiency failed to develop postoperative enteric gliosis, showed diminished immune cell infiltration, and were protected from POI. Furthermore, the IL1R1-deficiency in EGCs altered the surgery-induced glial activation state and reduced phagocytosis in macrophages, as well as their migration and accumulation around enteric ganglia. In patients, bowel surgery also induced IL-1-signaling, key molecules of enteric gliosis, and macrophage activation. Together, our data show that IL1R1-signaling triggers enteric gliosis, which results in *ME*-Mac activation and the development of POI. Intervention in this pathway might be a useful prophylactic strategy in preventing such motility disorders and gut inflammation.

[1] Department of Surgery, University Hospital Bonn, Bonn, Germany. [2] Department of Anesthesiology, Wexner Medical Center, The Ohio State University, Columbus, OH, USA. [3]These authors contributed equally: Fievos L. Christofi, Sven Wehner. ✉email: Sven.Wehner@ukbonn.de

A bdominal surgery induces an acute intestinal inflammation within the *muscularis externa* (*ME*)[1], resulting in functional motility disturbances, clinically known as postoperative ileus (POI). POI occurs in up to 27% of patients undergoing abdominal surgery[2] and is associated with prolonged hospitalizations, increased morbidity, and a high medico-economic burden[3]. As a result, patients suffer from nausea, vomiting, increased inflammatory response, and a higher risk of anastomotic leakage after colorectal surgery[4].

Studies of the past decades demonstrated that resident muscularis macrophages (*ME*-Macs)[5] as well as enteric glial cells (EGCs)[6] are key players and early responders in the postoperative *ME* inflammation[5,7,8]. In addition, both cell types lie in close anatomical association with enteric neurons, and there is growing evidence of communication between these cell types in health and gastrointestinal diseases in the context of inflammation[9–11].

EGCs express markers including S100β, glial-fibrillary acid protein (GFAP), proteolipid-protein-1 (PLP-1), or Sox10, commonly used as glial biomarkers to identify these cells throughout the intestine[11]. EGCs are present along the entire gastrointestinal tract, are known to modulate motility[12], and contribute to neuroinflammation in the gut[13]. Notably, some studies showed that EGCs maintain gut homeostasis[14] and that deletion of EGCs driven by the GFAP-promoter led to a massive inflammatory reaction in the GI tract[15,16], indicating a crucial immuneregulatory role of EGCs in the gut. However, these findings were challenged by a recent study showing that a proteolipid-protein-1 (PLP-1) driven depletion of EGC did not affect barrier maintenance nor sensitize mice to intestinal inflammation[12]. Despite controversial findings, EGCs are still discussed as promising interventional targets in several GI diseases, including POI[13,17]. EGC presence in the mucosa[18] and the *ME*[19] requires, in turn, the presence of intestinal microbiota. Furthermore, EGCs release immune mediators like interleukin-6 (IL-6) after stimulation with innate immune stimuli, including bacterial lipopolysaccharides[13], and host-derived factors like interleukin-1 (IL-1)[11]. Our group recently demonstrated that EGCs acquire a reactive phenotype after extracellular ATP treatment[20]. That study was the first to define the molecular identity of a reactive EGC phenotype, which we named "enteric gliosis". The general term *gliosis* is a well-established part of posttraumatic injury in the CNS[21] that involves intercellular communication between classical CNS immune cells, e.g., microglia, and neural cell types, including neurons and astrocytes. Based on similar functions and transcriptional profiles[22–25], *ME*-Macs and EGCs are often compared to microglia and astrocytes, respectively, and the bidirectional communication of the latter is a well-defined mechanism determining the functional fate of both cell types during inflammation[26].

In the intestine, there is growing evidence for an interaction of *ME*-Macs and cells of the enteric nervous system. Cell-to-cell communication between enteric neurons and *ME*-Macs has been shown to involve the release of colony-stimulating factor 1 (CSF1) and bone morphogenetic protein 2, respectively, and this interaction is thought to fine-tune intestinal motility[27]. Interactions between EGCs and *ME*-Macs were also recently described, showing that EGCs are a more potent source of CSF1. EGCs modulate visceral sensitivity through mechanisms that require interleukin-1β (IL-1β) and connexin-43 hemichannels (Cx43) to release CSF1, to activate *ME*-Macs[28]. While this study was the first to provide evidence on EGC–*ME*-Macs interactions in chronic intestinal inflammation, proof for this interaction in acute inflammation, occurring during POI, remains elusive. It is also unknown whether there is a link between EGC–*ME*-Macs interactions and abnormal motility in an acute inflammatory motility disorder such as POI.

We previously demonstrated that IL-1 is an essential cytokine in POI development, and pharmacological IL-1-antagonism with the drug anakinra or a global genetic IL1R1 depletion was shown to protect mice from POI[6]. Furthermore, we elucidated the role of the two ligands, IL-1 and IL-1β, in POI with a particular focus on IL-1β[29]. EGCs express IL1R1, and IL-1 stimulation results in the release of IL-6 and CCL2, which are known to play a role in POI[30–32]. Recently, IL-1-signaling in EGCs was also suspected of exerting pro-tumorigenic functions[33]. However, these studies demonstrated IL-1-induced cytokine release by EGCs only by carrying out in vitro experiments. Moreover, an EGC-specific approach, i.e., by cell-specific depletion of IL1R1 in EGCs, has not been used so far. Therefore, solid evidence of the cell-specific role of IL-1-signaling and focused analyses of EGC reactivity in vivo are still missing.

Herein, we aimed to fill this gap using two transgenic mouse models. First, the transgenic *RiboTag* mouse model allows a comprehensive in vivo "snapshot" of actively transcribed mRNA in EGCs[34]. Secondly, we generated a mouse with a targeted depletion of IL1R1 in glial-fibrillary acid (GFAP) expressing cells, one exclusive marker of EGCs. Additionally, we investigated if and how IL-1-signaling in EGCs affects *ME*-Mac responses by a series of transcriptional and functional analyses. Finally, we aimed to confirm our findings from our mouse study in human bowel specimens taken during a duodenopancreatectomy and in primary human EGC cultures isolated from GI surgical specimens. Together, our results show that IL-1 induces stimulus-specific enteric gliosis in EGCs that affects *ME*-Mac function and accounts for POI development in mice. Furthermore, findings in mice are translatable to humans since we discovered that the same molecular pathways are activated in the *ME* of the human postoperative bowel.

## Results

**IL-1 induces a specific type of gliosis in enteric glial cells.** Recently, we showed that surgical trauma and intestinal manipulation induce a reactive enteric glial cell (EGC) phenotype, also referred to as "enteric gliosis", coinciding with the release of various cytokines and chemokines[20]. We hypothesized that IL-1 triggers reactive gliosis, so we stimulated primary murine EGC cultures (Fig. S1a) with IL-1β and confirmed gliosis induction by validating known biomarkers that serve as hallmarks of a gliotic phenotype. IL-1β induced IL-6 release (Fig. S1b), proliferation (Fig. S1c), and changes in morphology (Fig. S1d) in EGCs. To generate a more comprehensive molecular signature profile of the IL-1 triggered alterations in gene expression in EGCs, we performed a bulk-RNA-Seq analysis. A principal component analysis (PCA) indicated a clear separation between IL-1β and vehicle-treated EGCs (Fig. 1a). Upon IL-1β treatment, 496 genes were up, and 826 were downregulated (Fig. 1b, $p < 0.05$; fold-change 1.5). Among the 20 most highly induced genes, we found inflammatory mediators like *IL-6*, chemokines (*Cxcl2*, *Cxcl5*, *Ccl2*, *Ccl5*), and colony-stimulating factors (*Csf1*, *Csf3*) (Fig. 1b). In addition, heat map analyses of genes sorted for induced inflammatory mediators confirmed these inductions (Fig. S1e). Next, we specified the reactive EGC state by gene ontology (GO) analysis, which revealed a gene enrichment within the terms "glial activation", "IL-1-signaling", "migration", and "inflammatory response" (Fig. 1c). Concurring heat maps confirmed a clear pattern in IL-1-activated EGCs (Fig. 1d + e), underlining their reactive state. Furthermore, by implementing our previously established "enteric gliosis" gene panel identified by ATP stimulation[20], we found a unique gene expression induced by IL-1β in EGCs, vastly different from our previously published ATP-induced gene panel (Fig. S1f). Comparison of our data set to published data investigating various stimuli for EGC activation, i.e.,

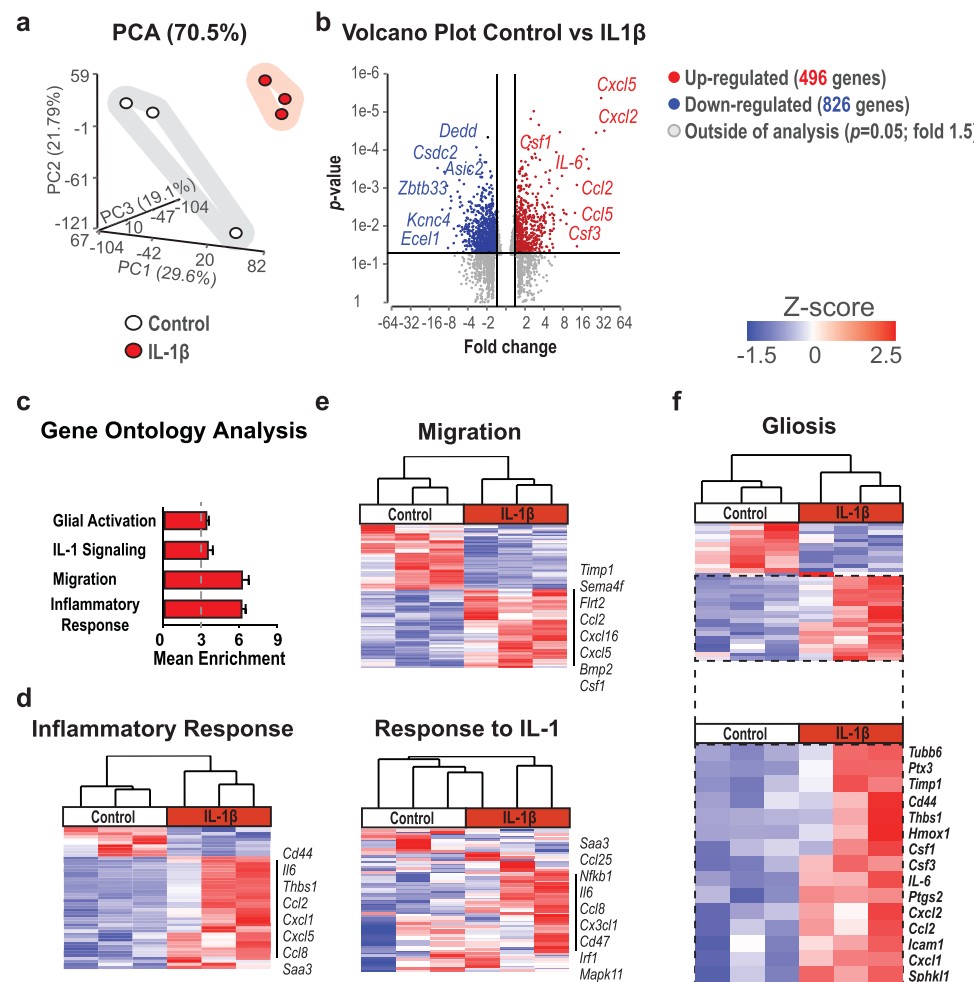

**Fig. 1 IL-1 induces a specific reactive phenotype in EGCs.** EGCs were treated with IL-1β (10 ng/ml) or vehicle (control) for 24 h and processed for bulk-RNA-Seq. **a** PCA plot shows a clear separation between both groups. **b** Volcano plot showing all significantly regulated genes. **c** Gene ontology (GO) analysis of IL1β-treated EGCs showing enrichment of genes connected to the indicated GO terms. **d**, **e** Heat maps of differentially expressed genes involved in "inflammatory response", "migration", and "response to IL-1" between IL-1β- and vehicle-treated EGCs. **f** Heat map of genes involved in "gliosis" highlighting the induced gene panel. Statistics were done with Fisher's exact test, n = 3 per group.

infection[35] or DNBS colitis[36], also revealed a distinct pattern for the IL-1β induced enteric gliosis phenotype with an inevitable overlap in key genes (Fig. S1g and Table S5). This analysis indicates that gliosis and EGC reactivity greatly depends on the inducing immune stimulus. IL-1-treated EGCs showed a prominent elevation of chemokine and cytokine levels involved in gliotic processes. Moreover, two new essential factors were identified in reactive glia in POI, *Csf1* and *Csf3* (Fig. 1f), with Csf1 recently implicated in EGC responses in a murine colitis model and human Crohn's Disease[28].

We conclude that IL-1 induces a specific transcriptome activation signature in EGCs, resulting in a distinct form of EGC gliosis leading to dysmotility and POI, with possible functions in inflammatory processes in the gut.

**IL-1-induced enteric gliosis resembles the EGC reactivity profile after surgical trauma.** In order to prove that the before-mentioned most prominent markers of the IL-1β-induced gliosis (see Fig. 1) are indeed selectively expressed by EGCs and not by other cell types present at low levels in EGC primary cell cultures, we used a Cre-recombinase-driven approach to analyze the expression of actively transcribed key gliosis genes. This approach is based on Cre-mediated (*Sox10^CreERT2*) tagging of the ribosomal Rpl22 protein with a hemagglutinin (HA) tag and subsequent

immunoprecipitation of the ribosomes, including actively transcribed mRNA (Fig. 2a and ref. [34]). Immunofluorescence microscopy confirmed selective expression of the HA tag in SOX10+ EGCs in cell cultures in vitro and, importantly, in vivo within *ME* whole mounts (Fig. 2b). Furthermore, after IL-1β treatment, key signature genes of gliosis (*GFAP*, *Nestin*) together with cytokines (*IL-6*, *TNFα*), chemokines (*Ccl2*, *Ccl5*, *Cxcl2*, *Cxcl5*), and growth factors (*Csf1*, *Csf3*) were selectively expressed and induced in EGCs (Figs. 2c and S2a).

As previous work implicated an immune-modulating role of EGCs in the postoperative inflammation of the *ME* resulting in POI[6,20], we next subjected *Sox10^CreERT2+xRpl22^HA/+* mice to intestinal surgical manipulation (IM) to investigate the reactive state of EGCs in vivo (Figs. 2d and S2b). As expected, 24 h after IM, these mice exhibited a strong postoperative infiltration of blood-derived leukocytes within the *ME* (Fig. S2c) and impaired GI-transit time (Fig. S2d), confirming regular induction of POI. Since we expected an early activation of EGCs after surgery, we then analyzed another group of the *Sox10^CreERT2+xRpl22^HA/+* mice at an early postoperative time point (IM3h). Bulk-RNA-Seq analyses of HA-precipitated EGC-specific mRNA revealed significant gene enrichment within the terms "IL-1-signaling", "migration", and "inflammation" (Fig. 2e). Heat maps for these GO-terms confirmed the activation pattern for an inflammatory

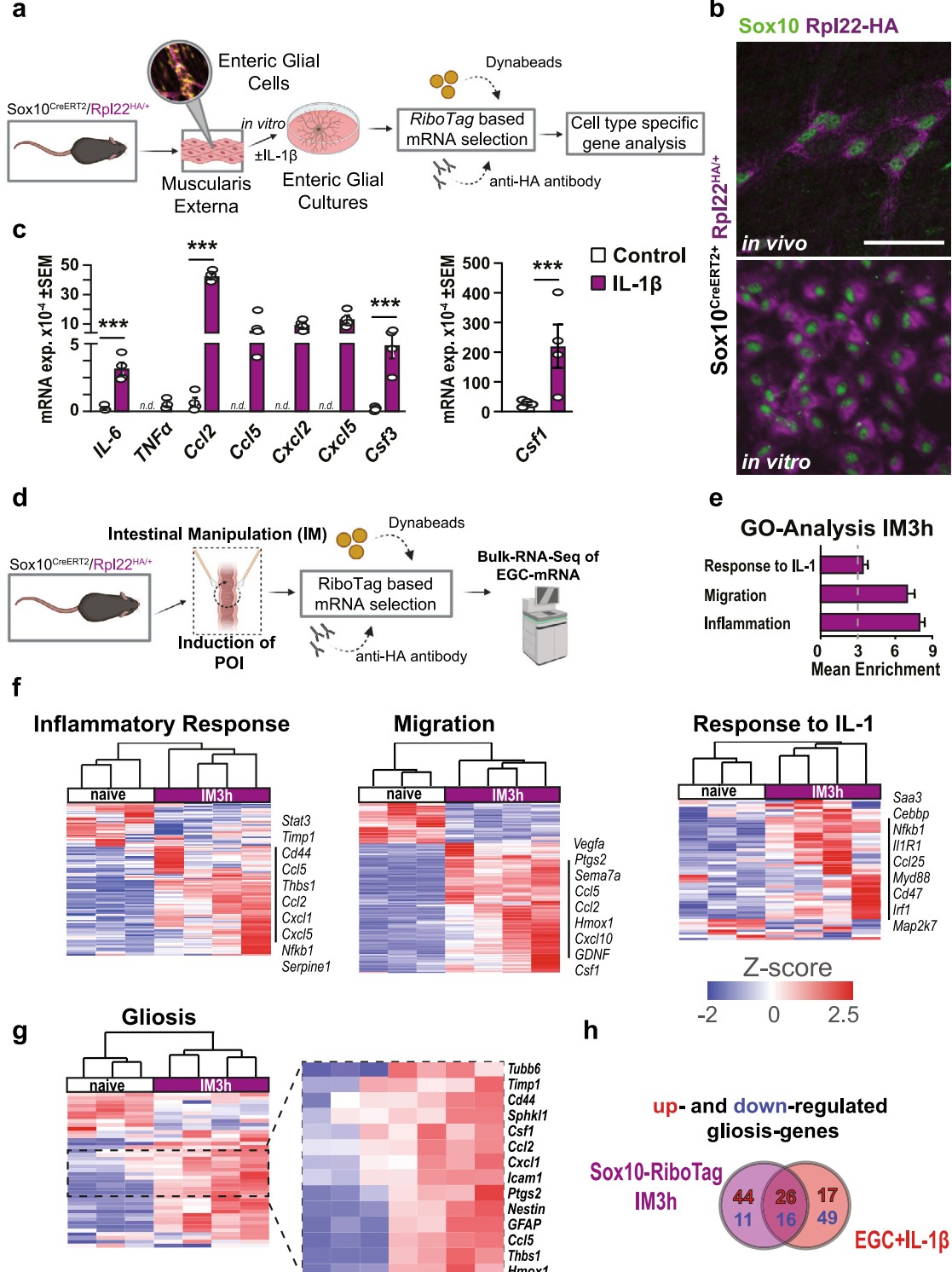

**Fig. 2 IL-1-induced enteric gliosis resembles the EGC reactivity profile after surgical trauma. a** Schematic overview of the *RiboTag* method applied in primary EGCs treated with IL-1β (10 ng/ml) or vehicle (control) for 3 h and processed for qPCR. *n* = 4 per group. **b** Representative immunofluorescence microscopy for EGCs (SOX10⁺, green) and the *RiboTag* (HA⁺, violet) in vivo (small intestine *ME*) and in vitro (EGC cultures) from *Sox10^{CreERT2+}-Rpl22^{HA/+}* mice. Scale bar 50 μm. **c** qPCR analysis for gliosis-associated genes in primary EGCs treated with IL-1β. Bars showing mean gene expression compared to 18S. *n* = 4. **d** Schematic overview of the *RiboTag* method applied in the POI animal model. *Sox10^{CreERT2+}-Rpl22^{HA/+}* mice underwent intestinal manipulation to induce POI, and EGC-specific mRNA was immunoprecipitated and used for bulk-RNA-Seq. *n* = 3–4 per group. **e** GO analysis of enriched genes at IM3h. **f** Heat maps of genes involved in "inflammatory response", "response to IL-1", and "migration". **g** Heat map of genes involved in "gliosis" highlighting the induced gene panel. **h** Venn-diagram of differentially expressed gliosis genes in IL-1β-treated EGCs and *Sox10^{CreERT2+}-Rpl22^{HA/+}* mice at IM3h. Statistics were done with Student's *t*-test. ***<0.001, * were compared to untreated EGC cultures.

response, as well as genes involved in migration and IL-1 responsiveness (Fig. 2f). Applying our enteric gliosis gene panel, we discovered an overlap of multiple IL-1-induced and IM-induced genes in EGCs. A heat map visualized the shared, induced genes, including inflammatory mediators (e.g., *Ccl2*, *Ccl5*, *Cxcl1*), structural proteins (e.g., *GFAP*, *Nestin, Tubb6*), activation markers (e.g., *Cd44, Icam1, Hmox1*), and growth factors (*Csf1, Csf3*) (Fig. 2g). A Venn-diagram showed the exact transcriptional overlap in gliosis genes between IL-1β-stimulated EGCs and the reactive EGC phenotype induced by surgical trauma (Fig. 2h and Table S6). More than 20% of the up-and downregulated gliosis genes overlapped, revealing IL-1 as a substantial part of surgery-induced EGC reactivity in vivo.

**EGC-restricted IL1R1 deficiency prevents postoperative macrophage activation and protects mice from POI.** Given that IL-1 induces EGC reactivity in POI, we next analyzed the functional significance of IL-1-triggered EGC gliosis on POI development. We subjected mice with an EGC-restricted IL1R1-deficiency (GFAP$^{cre}$xIL1R1$^{fl/fl}$, Fig. S3a) and IL1R1 competent control mice (GFAP$^{WT}$xIL1R1$^{fl/fl}$) to IM (Fig. 3a). To validate IL1R1-deficiency, we analyzed GFAP$^{Cre}$+xAi14$^{fl/fl}$ mice by immunohistochemistry for SOX10 and tdtomato expression, which confirmed a 95% overlay between the glial marker and the transgenic signal, thereby validating *Gfap* as a suitable Cre-promotor for a reliable IL1R1-deficiency in EGCs (Fig. S3b). Next, we performed bulk-RNA-Seq analysis on IM3h *ME* of conditional KO mice and wild-type littermates. The PCA plot revealed a clear separation between both groups (Fig. 3b), with 287 upregulated and 437 downregulated genes in GFAP$^{cre}$x-IL1R1$^{fl/fl}$ mice (Fig. S3c). Genes listed under the GO-terms "glial activation", "IL-1-signaling", "migration", and "inflammatory response" were enriched and more expressed in IL1R1 competent mice (Fig. 3c). A heat map of all differentially expressed genes (*p* < 0.05) revealed the downregulation of distinct gliosis genes in GFAP$^{cre}$xIL1R1$^{fl/fl}$ mice at IM3h, e.g., chemokines (*Ccl2, Cxcl1, Cxcl2, Ccl7*) and structural proteins (*GFAP, Cx43, Tubb6*) (Fig. 3d). To validate if these transcriptional alterations also affect the postoperative inflammation and the clinical outcome of POI (i.e., slow GI transit), we quantified the postoperative GI-transit and leukocyte infiltration 24 and 72 h after IM. Impressively, GFAP$^{cre}$+xIL1R1$^{fl/fl}$ mice were almost entirely protected from postoperative motility disturbances 24 h after IM (Fig. 3e), and leukocyte infiltration was severely reduced by more than 50% for myeloperoxidase$^{+}$ and 45% for CD45$^{+}$ leukocytes compared to GFAP$^{WT}$xIL1R1$^{fl/fl}$ mice (Figs. 3f and S3d). Gastrointestinal motility did not differ between genetically modified mouse strains (Fig. 3e) and C57BL6 wild-type mice upon surgery. Flow cytometry analyses confirmed reduced leukocyte infiltration by diminished numbers of F4/80$^{+}$ monocyte-derived macrophages, Ly6C$^{+}$ monocytes, activated (CD68$^{+}$) *ME*-Macs, and Ly6G$^{+}$ neutrophils in GFAP$^{cre}$xIL1R1$^{fl/fl}$ mice (Figs. 3g and S3d). Notably, confocal microscopy confirmed decreased numbers of MHCII$^{+}$ leukocytes around ENS ganglia in the postoperative *ME* of GFAP$^{cre}$+xIL1R1$^{fl/fl}$ mice (Fig. 3h), an effect probably based on the diminished expression of chemokines by EGCs. Indeed, a *NanoString* inflammatory panel revealed a severely diminished gene expression of inflammatory mediators (e.g., *Ccl2, Cxcl2, Cxcl5*), cytokines (e.g., *IL-6*), and a macrophage activation marker (*Arg1*) in the postoperative *ME* of GFAP$^{cre}$+xIL1R1$^{fl/fl}$ mice (Fig. 3i). In line, the postoperative increase of immune cell activation markers (*CD68, Mip-1a, Csf1, Csf3*), as well as gliosis markers (*IL-6, Ccl2, Nestin*, and *GFAP* was abrogated in GFAP$^{cre}$+xIL1R1$^{fl/fl}$ mice (Figs. 3j and S3e). Notably, similar results were observed in GFAP$^{cre}$xMyd88$^{fl/fl}$ mice, with an EGC-restricted

Myd88-deficiency, an essential adaptor molecule in the IL1R1 pathway. GFAP$^{cre}$+xMyd88$^{fl/fl}$ mice showed comparable down-regulated gene expression patterns as GFAP$^{cre}$+xIL1R1$^{fl/fl}$ mice and were protected from POI (Fig. S3f + g).

Based on these data, we conclude that an IL-1-mediated macrophage-glial interaction might be critical in POI development.

**Intestinal organotypic cultures demonstrate IL-1-dependent involvement in ME-Mac-EGC interactions.** Previously, we and others have shown that resident *ME*-Macs also play a critical role in POI. The close anatomical relation between EGCs and resident *ME*-Macs and the multiple immune mediators released by EGCs that can activate macrophages indicate that these cells communicate with each other, and EGCs might influence *ME*-Mac activation. However, upon surgical manipulation, inflammation attracts monocyte-derived macrophages that can hardly be distinguished from resident *ME*-Macs. In order to focus exclusively on the latter's response, we established an in vitro intestinal organotypic culture (IOC) model that closely reflects the resident in vivo cell composition and structures without being compromised by infiltrating immune cells. An aseptic separation procedure of the mouse jejunum *ME* was used to simulate the mechanical activation after surgical trauma. Separated *ME* specimens were either directly used as controls or cultivated for 3 h, reflecting the IM3h time point of the in vivo model (Fig. 4a). Notably, the separation procedure produced a stronger immune response in the *ME* than the gentle in vivo IM, which was indicated by the increased gene expression of gliosis genes in IOCs than in vivo manipulated ME after 3 h (Fig. S4a). Immunofluorescence microscopy for the activation marker FOSb as well as SOX10 and β-3-tubulin (TUBB3) showed intact and activated FOSb$^{+}$ EGCs after 3 h incubation but no activation in control *ME* (Fig. S4b). Strong *Fosb* induction was confirmed by qPCR (Fig. S4c). Interestingly, after the 3 h incubation period, we counted significantly more activated CD68$^{+}$ *ME*-Macs around enteric ganglia (Figs. 4b + c and S4d). Moreover, 3D reconstruction identified the near localization of CD68$^{+}$ *ME*-Macs around enteric ganglia (Fig. S4e). In line with our in vivo POI mouse data, qPCR measurements of established markers for gliosis (*Nestin, GFAP*), intestinal inflammation (*IL-6, IL-1β, Ccl2*), and immune cell activation (*Mip-1a, CD68, Csf1, Csf3*), showed significant induction in the 3 h cultured IOCs (Fig. 4d). We transferred this model also to IOCs generated from GFAP$^{cre}$+xIL1R1$^{fl/fl}$ and their Cre-negative littermates. The former showed a weaker induction of gliosis, intestinal inflammation, and immune cell activation genes (Fig. 4e), indicating that the absence of EGC-specific IL-1-signaling reduces the trauma-induced *ME*-Mac activation.

**The impact of EGC-derived factors on macrophage function.** To further elucidate the interactions of EGCs and *ME*-Macs, we treated primary EGCs with IL-1β or vehicle and transferred the conditioned media (CM) to IL1R1-deficient bone-marrow-derived macrophage (BMDM) cultures (Fig. 5a). The primary EGC cultures consist of at least 80% glia and low amounts of neurons (Fig. S5a) and fibroblasts, verified by ICC[20] and RNA-Seq analysis for relevant cell type markers (Fig. S5b). To verify the IL-1-induced gliosis and routinely perform quality control of the CMs, we measured protein levels of prominent inflammatory markers, IL-6 and CCL2 (Fig. S5c). Moreover, to confirm that only the EGCs in our primary cultures react to the IL-1-activation by releasing inflammatory mediators, we generated primary EGCs from GFAP$^{Cre}$+xIL1R1$^{fl/fl}$ and GFAP$^{WT}$xIL1R1$^{fl/fl}$ mice. GFAP$^{Cre}$+ EGC cultures showed lower expression of inflammatory mediators on protein (Fig. S5d) and mRNA (Fig. S5e) levels,

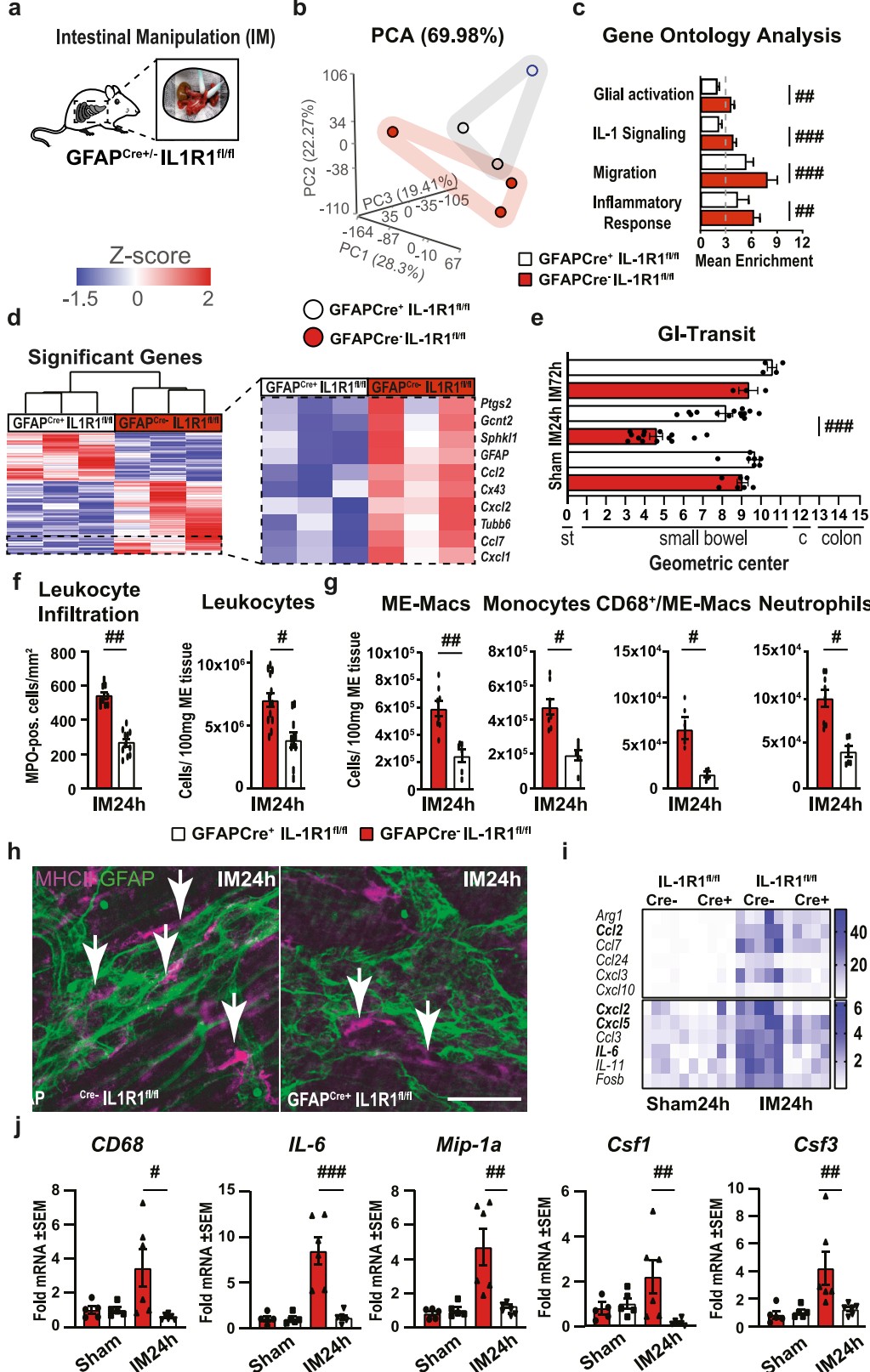

proving that these factors are released by EGCs into the CM after IL-1 treatment (CM^{IL-1}).

After 3 h treatment with the EGC-conditioned media, BMDMs were analyzed by bulk-RNA-Seq. The resulting PCA demonstrated a clear separation of the groups (Fig. 5b). Notably, we already observed altered functionality between naïve BMDMs and

those treated with CM of vehicle-treated EGCs (CM^{Veh}). This effect was even more pronounced in BMDMs treated with the CM^{IL-1} (Fig. S5f), resulting in 762 up- and 1656 downregulated genes (Fig. 5c). Within the list of differentially expressed genes, classical M1 (*Il1b*, *Tlr4*, *Sphk1*) and M2 (*Arg1*, *Socs3*, *Tgfb1*) markers showed that IL-1-triggered EGC factors affect the

**Fig. 3 EGC-restricted IL1R1 deficiency prevents postoperative macrophage activation and protects mice from POI. a–d** GFAP[Cre+]-IL1R1[fl/fl] and GFAP[Cre-]-IL1R1[fl/fl] mice underwent intestinal manipulation (IM) to induce POI. After 3 h, RNA of the ME was processed for bulk-RNA-Seq. $n = 3$ per group. **b** PCA plot of Cre+ and Cre− groups. **c** GO enrichment analysis at IM3h comparing Cre+ and Cre− animals, normalized to respective sham mice. **d** Heat map of differentially expressed genes highlighting induced gliosis genes. **e–j** GFAP[Cre+]-IL1R1[fl/fl] and GFAP[Cre-]-IL1R1[fl/fl] mice were analyzed at IM24h. **e** GI-transit analysis in IM72, IM24h, and sham24h groups of Cre+ and Cre− mice. $n = 7$ (Sham); 14 (IM24h); 3–4 (IM72h). **f** Leukocyte infiltration quantification by MPO histology and FACS in IM24h groups of Cre+ and Cre− mice. $n = 10$ per group. **g** FACS analysis for ME-Macs (CD45+, F4/80+), monocytes (CD45+, Ly6C−, Ly6G+), activated ME-Macs (CD68+, F4/80+), and neutrophils (CD45+, Ly6C+, Ly6G−) in IM24h groups of Cre+ and Cre− mice. $n = 5$ per group. **h** Representative immunofluorescence microscopy of ME whole mounts showing EGCs (GFAP+, green) and ME-Macs (MHCII+, violet) in IM24h groups of Cre+ and Cre− mice. Scale bar 50 μm. **i, j** Gene expression analysis by inflammatory NanoString panel (**i**) and by qPCR (**j**) for genes related to inflammation (IL-6) and immune cell activation (CD68, Mip-1a, Csf1, Csf3) at IM24h in Cre+ and Cre− mice, normalized to corresponding sham groups. $n = 5$ per group. Statistics were done with Student's t-test and Fisher's exact test. # < 0.05, ## < 0.01, ###<0.001, # were compared to Cre− littermates.

macrophage polarization status. GO analysis and heat maps displayed substantial changes in gene clusters for "migration", "phagocytosis", and "inflammatory response" (Fig. 5d + e). Consequently, we analyzed EGC-CM-treated BMDMs for migration and phagocytosis, with the FCS treatment as a positive control. In wound-healing and transwell assays, we observed increased migration of BMDMs treated with CM[IL-1] compared to control and CM[Veh] (Figs. 5f + g and S5g). Phagocytosis, measured by FITC-dextran uptake after 2 h, was also increased (Fig. 5h). Overall, our data indicate that EGCs already affect macrophage function and polarization under resting conditions; however, macrophage activation and functionality are far more altered by factors released from IL-1β-stimulated EGCs.

**Enteric gliosis and IL-1-signaling are involved in acute intestinal inflammation after abdominal surgery.** Finally, we validated the existence of IL-1-dependent enteric gliosis in humans. Histology on intraoperatively taken jejunal specimens from patients undergoing open abdominal surgery for pancreatic head resection confirmed the close association of CD68+ ME-Macs and GFAP+ EGCs in the ME (Fig. 6a) and a strong IL1R1 expression in EGCs (Fig. S6a + b). Notably, this oncological surgery allowed the collection of tumor-free bowel samples at an early and a late time point during surgery (Fig. 6b). By validating the enteric gliosis status in the patient samples, we discovered the induction of inflammatory mediators (IL-1α, IL-1β, IL-6, CCL2, CXCL2), immune cell activators (CSF1, CSF3, MIP-1a), and EGC gliosis markers (GFAP, NESTIN) (Figs. 6c and S6c), previously identified in our mouse studies. To expand our understanding of ongoing mechanisms in the tissue after surgery, we performed bulk-RNA-Seq on early and late intraoperatively taken jejunal ME specimens and found more than 400-differentially regulated genes (Fig. 6d + e). In line with our murine data sets, gene clusters associated with "glial activation", "IL-1-signaling", "migration", and "inflammatory response" were enriched in intestinal samples from late surgery time points (Fig. 6f), correlating with corresponding heat maps (Fig. S6d). Moreover, heat map visualization of significantly regulated genes ($p < 0.05$) revealed the upregulation of distinct gliosis genes, e.g., chemokines (CCL2, CXCL2), structural proteins (TUBB6), and activation markers (HMOX1, SOCS3, ICAM1) during surgery (Fig. 6g), indicating the manifestation of enteric gliosis during surgery. To validate the effect of IL-1 on surgical samples, we prepared human jejunal IOCs from late collected patient material and incubated them with the IL1R1-antagonist anakinra for 24 h, previously used in vivo in the POI model[6]. Anakinra-treated human IOCs showed lower expression of CCL2 and IL-6 (Fig. S6e), indicating a dampened inflammatory activation response. We finally tested if primary human EGC cultures also become reactive upon IL-1β-stimulation (Fig. 6h). We detected induction of inflammatory mediators such as IL-6, TNFα, CCL2,

CXCL2, and IL-8 on mRNA level (Fig. 6i) as well as IL-6, and CCL2 on protein level (Fig. S6f) after IL-1β treatment, underlining that the gliosis phenotype is conserved across species.

Together, our data demonstrate that enteric gliosis is also induced in patients and that IL-1 induces EGC reactivity and production of various mediators that can modulate ME-Mac activation and functionality, thereby guiding postoperative intestinal inflammation and POI.

Overall, our investigation demonstrates that IL-1 induces a reactive phenotype in mouse EGCs. This distinct enteric gliosis state is part of the postoperative immune cascade in the ME in both mice and humans, and an EGC-restricted IL1R1 deficiency ameliorates POI in mice. Furthermore, mediators released by reactive EGCs alter the phenotype and function of resident ME-Macs towards an activated macrophage type with stronger migratory and phagocytic capabilities.

## Discussion
Over the last 2 decades or so, the role of glial cells evolved from "giving structural support" to "issuing orders", especially after trauma and under inflammatory conditions, with a multitude of examples in the central[21] and enteric nervous system[11]. The present study advances our understanding of the molecular mechanisms in EGCs in an inflamed gut, and it provides an assessment of the communication between EGCs and macrophages during acute intestinal inflammation in POI. In particular, we demonstrate that EGCs acquired a reactive phenotype after surgical trauma and defined this reactive state as enteric gliosis. A key discovery is that IL1R1-signaling is a critical driver of EGC reactivity in this process. Furthermore, we found that reactive EGCs communicate with intestinal ME-Macs, whose cellular functions become altered towards an activated phenotype, characterized by increased migratory and phagocytic activity. Selective disruption of IL1R1-signaling in EGCs prevented reactive gliosis and protected mice from postoperative ME inflammation, motility disturbances, and POI.

Previously, we revealed that a pan-IL1R1 deficiency protected mice from postoperative bowel wall inflammation[6]. Our study added IL-1 signaling to a list of key factors in immune-mediated motility disorders that are used as an indicator for a successful therapy like the cytokine IL-6[30], which is also induced by IL-1[6,29], or TNFα, which is essential in the later stages of POI[37]. In POI development, various inflammatory mediators, like danger-associated molecular patterns (DAMPs: e.g., ATP[20] and IL-1α[29]) and pathogen-associated molecular patterns (PAMPs: e.g., dsRNA[7], and LPS[38]), play an essential role and contribute to the disease progression[39]. However, our recent work highlighted that IL1R1 is expressed on EGCs, and discovered that IL-1-signaling in EGCs might be a key component in intestinal inflammation and motility impairment after abdominal surgery[6].

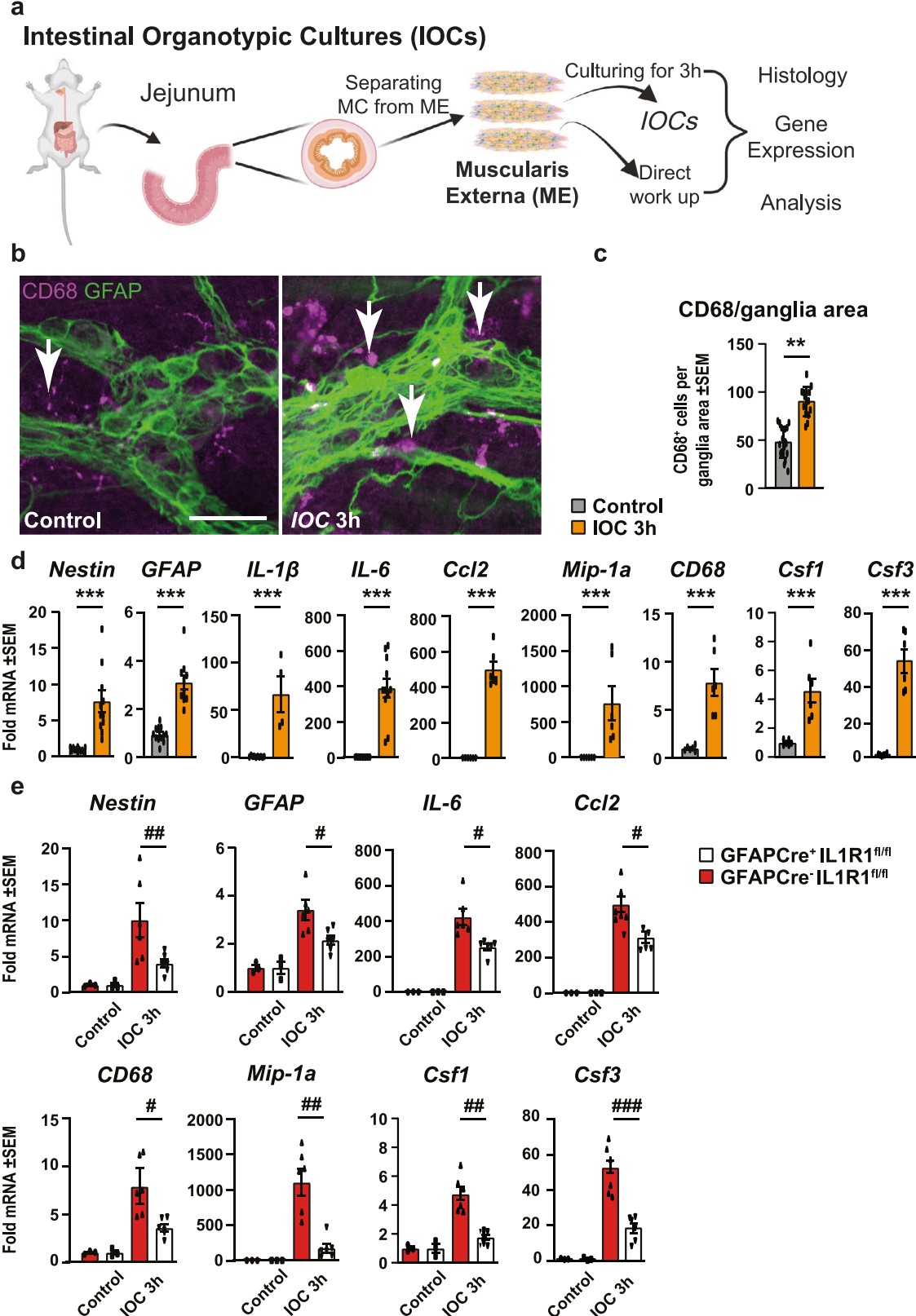

Herein, we now proved the importance of glial IL1R1-signaling with several transgenic mouse models that allowed us to precisely analyze and modify EGC reactivity in vivo and in vitro. One of these models is the $Sox10^{CreERT2+}xRpl22^{HA/+}$ (*RiboTag*) mouse, suitable for generating an in vivo "snapshot" of actively transcribed RNA selectively in EGCs[34,36]. Analysis of

*RiboTag* mice confirmed that IL1R1-signaling is part of post-operative EGC reactivity. Around 2/3 of upregulated genes overlap between IL-1β-stimulated EGCs in vitro and the reactive EGC phenotype induced by IM in vivo. We conclude that an IL-1-induced EGC reactivity is a major trigger of surgery-induced enteric gliosis.

**Fig. 4 Intestinal organotypic cultures demonstrate IL-1-dependent involvement in ME-Mac-EGC interactions. a** Experimental workflow scheme. Intestinal organotypic cultures (IOC) from *muscularis externa* (*ME*) were prepared from C57BL/6 wild-type mouse jejunum by surgical dissection of the *lamina propria* and mucosa tissue, a procedure that mimics the surgical trauma in vivo. This model allows the analysis of resident cell types in the absence of any infiltrating blood-derived immune cells. IOCs were either directly processed or incubated for 3 h, corresponding to the IM3h time point in the POI mouse model. $n = 6$ per group. **b, c** Histological visualization (**b**) and quantification (**c**) of activated macrophages (CD68+, violet) surrounding enteric ganglia (GFAP+, green). $n = 5$ mice per group; quantifying ganglia in 3–4 IOCs per mouse. Scale bar 50 µm. **d** Gene expression analysis in IOCs by qPCR for genes involved in "enteric gliosis" (*Nestin, GFAP*), inflammation (*IL-1β, IL-6, Ccl2*), and *immune cell activation* (*Mip-1a, CD68, Csf1, Csf3*). **e** IOCs were prepared from GFAP$^{Cre+}$-IL1R1$^{fl/fl}$ and GFAP$^{Cre-}$-IL1R1$^{fl/fl}$ mice according to the same experimental workflow shown in **a** and either directly processed or incubated for 3 h. Gene expression analysis by qPCR for genes involved in "enteric gliosis", "inflammation", and "immune cell activation". $n = 3–8$ per group. Statistics were done with Student's $t$-test. */#<0.05, **/##<0.01, ***/###<0.001, All * compared to control IOCs, all # compared to Cre$^-$ littermates.

Further evidence for stimulus-selective reactivity in EGCs comes from the comparison of IL-1β- and ATP-stimulated EGCs. We recently showed that ATP induces enteric gliosis[20]. Compared to ATP, IL-1β caused a stronger induction of several chemokines, suggesting a specific role in reactive processes upon surgery. These findings imply that the term "reactive glia", commonly used to describe a general glial activation, does not represent a hardwired phenotype as it depends on the specific stimulus. However, some genes were upregulated in ATP-and IL-1β-stimulated EGCs, including *IL-6, Cxcl1*, and *Cxcl2*, prominently induced in astrogliosis[40], and *Hmox1*, a marker for neurodegeneration[41] and reactive glial cells[42]. These genes might be part of an overarching response found in multiple reactive EGC phenotypes, pointing to a core gliosis signature across various organs and stimuli. Future comparisons of enteric gliosis signatures from other glial cell populations as well as other diseases might help to specify both the core gliosis signature and the stimulus- and disease-specific responses of reactive glial cells. Usage of published data sets from activated EGCs under inflammatory conditions (infection[35] and colitis[36]) showed that IL-1 induced gliosis shares induced genes and shows individual molecular responses. However, the available data were collected with different approaches, mouse models, intestinal parts, and transcriptional analysis methods, providing only a rough initial idea of core genes and conditions leading to specific gliosis expression patterns. More comprehensive comparisons with multiple conditions are required and expected to be published in the future to allow reliable conclusions on core and conditions-specific gliosis signatures.

Given that IL-1-signaling induced the expression of a variety of immune mediators, which might preferably act on immune cells located in anatomical proximity, we focused on the interactions of EGCs and *ME*-Macs. These *ME*-Macs have previously been shown to be crucial in the postoperative immune response in POI[5,37,43]. An ex vivo IOC model allowed us to explicitly focus on the EGCs and ME-Macs interaction. In manipulated IOCs, we found an upregulation of several molecules depending on glial IL1R1-signaling known to act on macrophages and affect their function. Among these genes, we found unspecific macrophage activation markers (*Mip1* and *CD68*) and members of the *Csf1* family (*Csf1* and *Csf3*), which exert distinct functions on macrophage differentiation.

Furthermore, CSF1 was described as an essential factor in maintaining *ME*-Macs[44]. Although enteric neurons were first identified as a CSF1 source in the intestine[27], a recent study by the Gulbransen group showed that EGCs produce more *Csf1* than enteric neurons in a colitis model[28]. They also showed that IL-1 triggers CSF1 release from EGC cultures in vitro *and* increases MHCII and CD68 expression in BMDM cultures. Our IOC model and in vivo data add to this hypothesis, showing that EGCs indeed produce CSF1 and activate *ME*-Macs in their native environment. Notably, the IOCs' initial immune response was

stronger than the in vivo manipulated ME response at 3 h after mechanical separation or surgical manipulation. We interpret this difference as a consequence of an increased release of additional factors due to the stronger mechanical forces applied to the IOCs during their preparation than the more gentle in vivo ME manipulation. These mediators might trigger additional immune responses independent of IL-1 release.

Consequently, IL1R1 depletion or antagonism does not dampen the cytokine production and EGC activation in the IOC model to the same extent as in the more gentle in vivo surgical manipulation approach. Here, it should be noted that the IOC does not fully reflect POI as it is missing important characteristics of POI development. First peripheral innervation of the SNS and PNS, known to modulate the ME immune response to surgery, is missing. Secondly, and more critical for the later inflammatory phase IOCs do not become infiltrated by blood-derived leukocytes, which extravasate into the *ME* after surgical manipulation. For these reasons, IOCs do not represent an ex vivo POI model but a suitable model for studying local tissue responses to mechanical or surgical trauma without systemic stimuli. Research using *ME* IOCs might be a tool to investigate how the tissue can immunologically respond in vitro to what might affect molecular and functional measurements, e.g., in pharmacological and physiological studies, respectively.

Another interesting molecule expressed by EGCs in an IL-1-dependent manner is *Csf3*. Compared to other members of the *Csf* family, *Csf3* is less well studied, and to date, it has been linked to neutrophil migration and activation[45,46]. As we also observed a prominent induction of the neutrophil chemokines *Cxcl2* and *Cxcl5* and a significant reduction of infiltrating neutrophils in GFAP$^{Cre}$xIL1R1$^{fl/fl}$ mice after surgery, EGCs might indeed affect the recruitment and maybe even the function of these cells. As neutrophils are not present at the beginning of intestinal surgery and instead infiltrate at later stages, an interaction of reactive EGCs with these infiltrating immune cell populations is likely and warrants future investigation. Importantly, recent studies showed that *Csf3* is able to alter macrophage polarization in vitro[47] and induce the expression of regulatory macrophage markers in DSS-colitis[48]. In breast cancer, *Csf3*-signaling also caused immunosuppressive behavior in macrophages[49], making *Csf3*, next to *Csf1*, one of the most promising EGC-released factors that impact the state and function of *ME*-Macs and/or infiltrating monocyte-derived macrophages.

EGC-selective IL1R1-deficiency completely abolished the expression of all the mediators directly acting on macrophages (i.e., Csf1, Csf3, Cxcl1, Cxcl2, Il6, Ccl2, and Mip1α), we expected that this would also alter macrophage transcriptomes and function. The capacity of EGCs to induce inflammatory responses in macrophages was indeed confirmed by RNA-Seq analysis showing changes in the inflammatory response and the expression of M1 and M2 markers. In addition to the general induction of immune responses, genes shaping essential macrophage functions

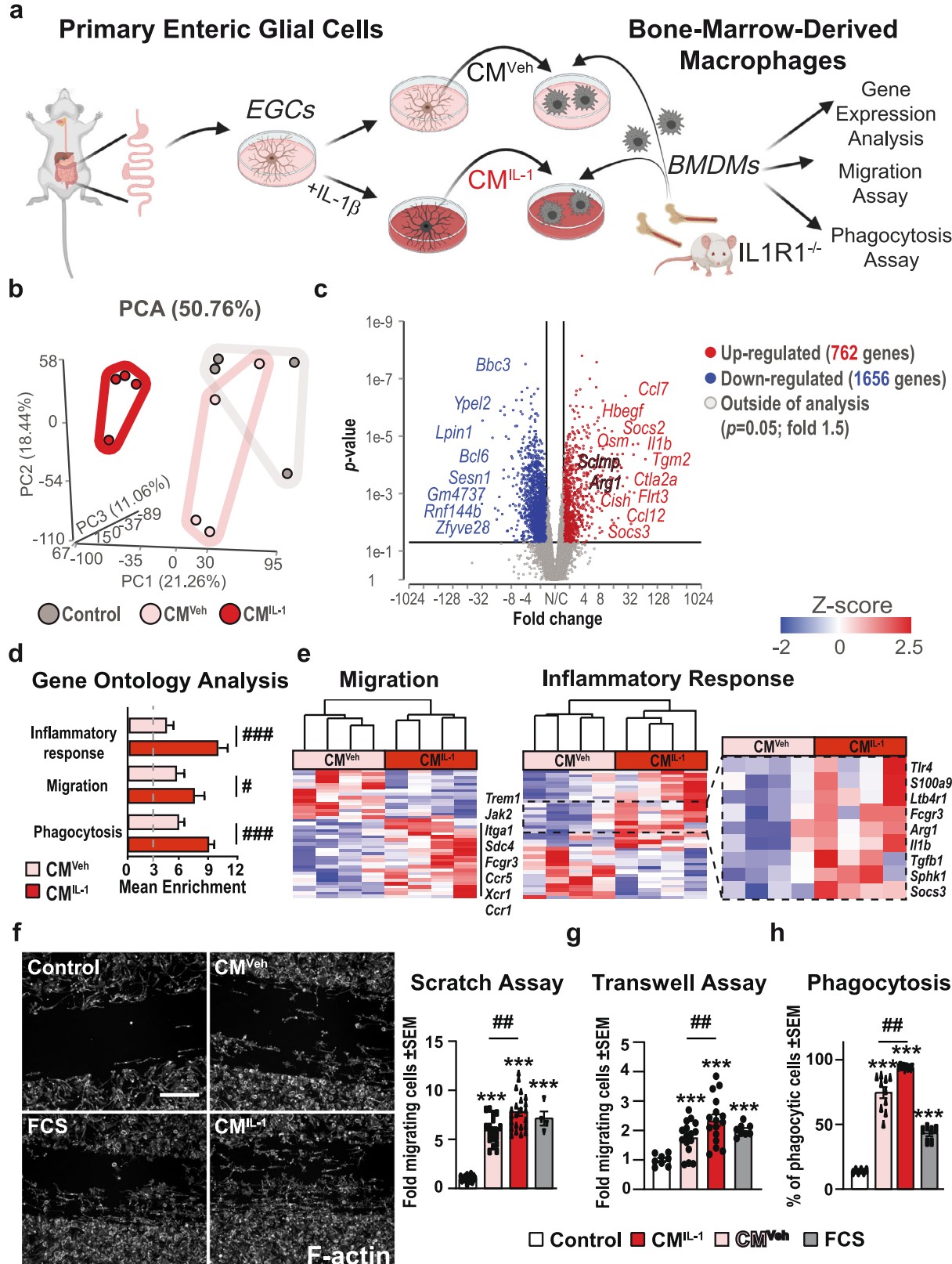

like migration and phagocytosis were also transcriptionally altered in response to reactive EGC-released factors. The accumulation of CD68[+] *ME*-Macs around enteric ganglia and their morphological changes towards a round-shaped cell type showed that *ME*-Mac migration and activation are increased upon exposure to EGC-derived mediators. Similar to microglia in the

CNS, the attraction of *ME*-Macs to the vicinity of enteric ganglia might be part of a protective mechanism involving either rapid clearance of neuronal debris or neuroprotective features against excessive inflammatory responses to restore intestinal function. Increased resident macrophage accumulation at sites of neuronal damage is also observed in models of spinal cord[50] and brain

**Fig. 5 The impact of EGC-derived factors on macrophage function. a** Schematic overview showing the generation of primary EGCs and production of CM$^{IL-1}$ (EGCs treated with IL-1β (10 ng/ml) for 24 h) and CM$^{Veh}$ before transferring CM to Bone-marrow-derived macrophages (BMDMs). BMDMs were isolated from IL1R1$^{-/-}$ mice to exclude any side effects of IL1β-residues in the collected CM$^{IL-1}$. After BMDM maturation, cells were treated with CM$^{IL-1}$, CM$^{Veh}$, or were left untreated for 3 h (RNA-Seq) and 24 h (functional assays). **b** Bulk-RNA-Seq analyses of BMDMs after 3 h incubation showed a separation between the three treatment groups in a PCA plot. $n = 4$ per group. **c** Volcano plot from bulk-RNA-Seq analyses of BMDMs after 3 h treatment with CM$^{IL-1}$ or CM$^{Veh}$ highlighting the top 20 regulated genes (Fold 2, $p$-value 0.05). **d, e** GO enrichment analysis in BMDMs after CM$^{IL-1}$ and CM$^{Veh}$ stimulation and heat maps of genes involved in "migration" and "inflammatory response" highlighting induced genes related to macrophage function. **f–h** BMDMs were processed 24 h after CM incubation in **f** scratch ($n = 4$ independent BMDM cultures; multiple technical replicates per culture; Control (20 readings); CM$^{Veh}$ (20 readings); CM$^{IL-1}$ (20 readings); FCS (4 readings). **g** transwell ($n = 3$ independent BMDM cultures; multiple technical replicates per culture; Control (8 readings); CM$^{Veh}$ (16 readings); CM$^{IL-1}$ (16 readings); FCS (8 readings), and **h** phagocytosis ($n = 3$ independent BMDM cultures; multiple technical replicates per culture; Control (6 readings); CM$^{Veh}$ (10 readings); CM$^{IL-1}$ (10 readings); FCS (6 readings) assays. Scale bar 100 μm. Statistics were done with Student's $t$-test and Fisher's exact test. */#<0.05, **/##<0.01, ***/###<0.001, All * compared to control BMDMs, all # compared to CM$^{Veh}$-treated BMDMs.

injury[51,52], wherein microglia, the CNS counterparts of ME-Macs, accumulate in the damaged areas. Support for increased clearance of dying cells or debris comes from our observation that IL-1-triggered EGCs stimulate phagocytosis in macrophages. As ENS homeostasis is well-ordered by apoptosis and neurogenesis[53], an enhanced elimination of damaged neurons or their protection from excessive inflammation by phagocytosis of cell debris might be a conserved mechanism. Indeed, a neuroprotective role of ME-Macs during homeostasis[22] and infection has recently been shown[54]. When enteric gliosis is a key element of this neuro-protective response, a logical argumentation would be that its blockage might have detrimental effects. However, as we observed a clear improvement of POI and a reduction of postoperative inflammation by inhibiting IL1R1-mediated enteric gliosis, we suppose that neuroprotective mechanisms might play a minor role during acute postoperative inflammation and that compensatory pathways are activated to maintain this function. Another possibility is that ME-Mac migration to ganglia might be fundamentally a protective homeostatic mechanism, although a stronger reactive glial cell phenotype with more surgical trauma may cause an exaggerated response and activation of macrophages to act instead in a detrimental way to exacerbate the inflammatory response. Our previous work[5] and other investigators[31] have provided evidence that activated macrophages are essential contributors to the development of POI. We propose that glial IL1R1 knockout may protect mice from POI by causing a milder macrophage induction in these mice. However, a causative link between glia-to-macrophages-to neurons has not yet been established in our study for the glial IL1R1 knockout model and the resolution of the GI-transit phenotype in the POI model in GFAP$^{Cre}$xIL1R1$^{fl/fl}$ animals does not necessarily depend on a milder macrophage induction and glial-to-macrophage signaling. An alternative hypothesis to be tested is that IL1-induced EGCs directly influence enteric neurons that control peristalsis to cause POI and that a glial IL1R1 knockout prevents it. Recent studies show that EGCs can interact with enteric neurons, triggering cell death[55,56], neuronal dysregulation[57], and homeostasis functions[58]. These interesting questions need to be addressed in future studies. Notably, different subtypes of ME-Macs with different innate responsiveness have recently been described by us[7], and previous single-cell-RNA-Seq studies revealed four distinct clusters of ME-Macs[22]. Unfortunately, the individual subpopulation-specific cellular functions and interactions are largely unknown, and future studies have to investigate whether EGCs preferably interact with one of these distinct subtypes.

As already stated above, besides interacting with resident ME-Macs, EGCs might also act on infiltrating monocytes, which are non-exclusively attracted by EGC-released CCL2. CCL2 is a major chemoattractant for monocyte-derived leukocytes in POI[32,59]. Although three groups independently demonstrated that these cells

do not affect POI 24 h after surgery, they are involved in the late-phase resolution, and CCR2-deficient mice (i.e., the receptor target for CCL2) recover more slowly from POI than wild-type animals[31,32,59]. This observation might point towards a potential beneficial role of EGC-derived CCL2 in POI resolution. However, in our view, this seems unlikely as we already observed a clear improvement of POI after 24 h and normalization of GI-transit after 72 h in GFAP$^{Cre+}$-IL1R1$^{fl/fl}$ mice. Hence, it is evident that there is no delay in POI at later stages.

The human data included in our study showed evidence that IL-1-signaling and enteric gliosis also occur during GI-tract surgery. Although the time between sampling of the early and late intraoperative specimen was less than 3 h, the earliest time point in our mouse studies, we observed a strong induction of cytokines and chemokines comparable to murine IL-1-triggered enteric gliosis. Notably, IL-1β was also strongly induced, and human EGCs had a strong transcriptional response, indicating a preserved mechanism of enteric gliosis in rodents and humans. Moreover, the GO analyses underlined the induction of common glial activation and immune-activating pathways. The data is translatable from mice to humans, and we conclude that the blockade of IL-1-triggered enteric gliosis might therefore prevent the development of POI in surgical patients. Previous preclinical data from our group already demonstrated that an antibody-mediated depletion of IL-1α or IL-1β or pharmacological inhibition of IL1R1 by Anakinra effectively prevented POI in mice[6]. Our findings in human IOCs, isolated from surgical patients, showing reduced IL-6 and CCL2 upon ex vivo stimulation in the presence of Anakinra, confirm our theory about an immediate EGC immune-responsiveness to IL-1 signaling. These findings emphasize our view that the initial inflammatory activation of the EGC/ME-Mac axis is crucial for POI development, and interaction with IL-1-signaling might be rather useful for prevention than treatment of existing POI. In line, multiple trials with Anakinra show anti-inflammatory effects in inflammation-associated intestinal diseases, such as mucositis (NCT03233776) and colorectal cancer (NCT02090101). Anakinra or other pharmacological interventions targeting IL1-signaling in EGCs could potentially serve as a prophylactic treatment in POI in surgical patients to add to the benefits of ERAS (enhanced recovery after surgery) protocols[60].

In conclusion, our study provides insights into the molecular mechanism of postoperative IL-1-triggered enteric gliosis and its consequence on the communication between EGCs and ME-Macs. Inhibition of this gliotic state ameliorated postoperative inflammation and protected mice from POI. Moreover, we confirmed the induction of enteric gliosis and activation of IL-1-signaling in surgical patients, supporting the idea of an intervention in the IL-1 pathway as a promising and clinically suitable strategy to prevent inflammation-induced motility disorders.

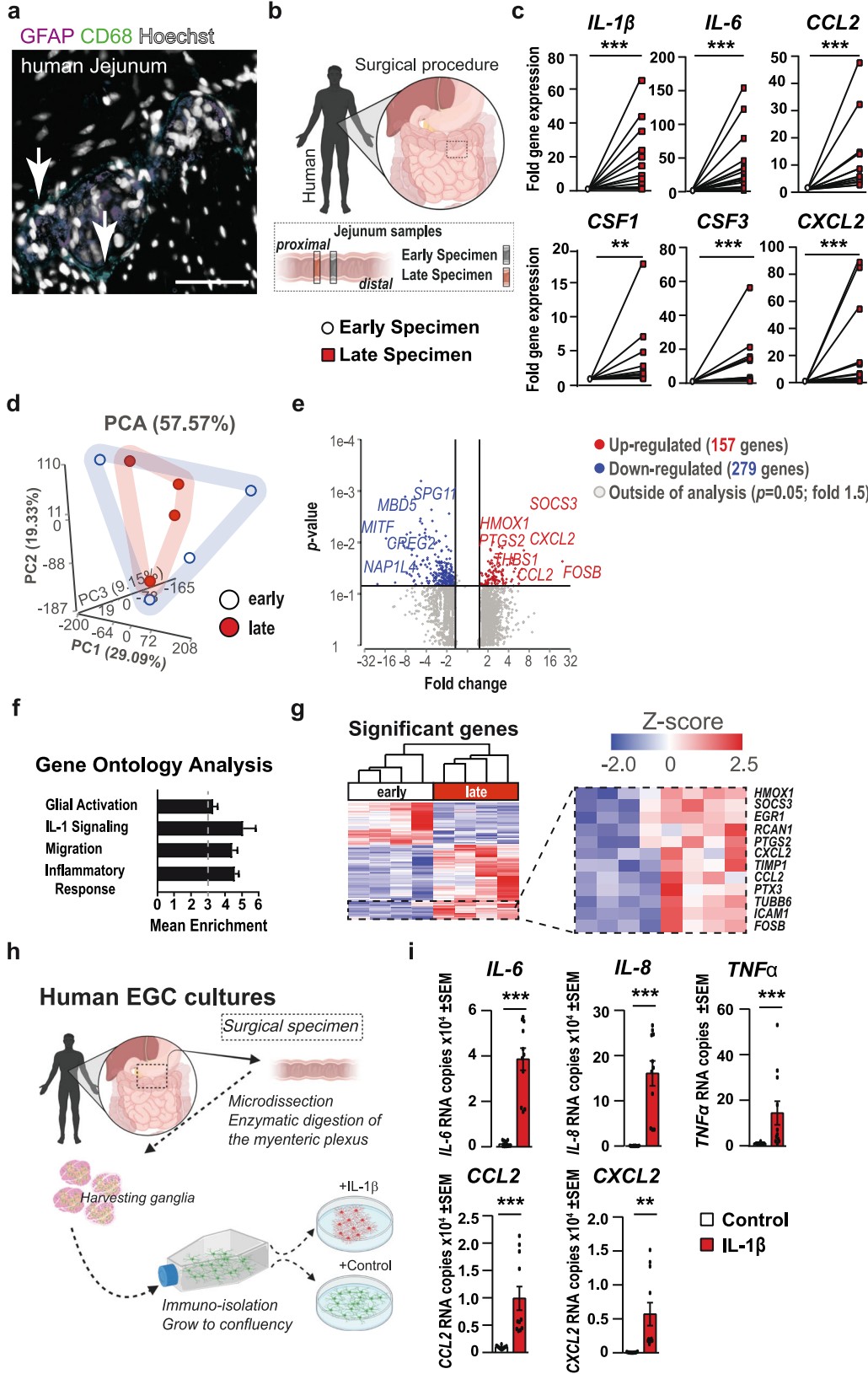

## Methods

**Animals**. Experiments were performed using 8–12-week-old Sox10-CreERT2xRpl22HA/+ or GFAPcrexIL1R1fl/fl mice kept in a pathogen-free animal facility with standard rodent food and tap water ad libitum. Appropriate authorities of North-Rhine-Westphalia, Germany (81-02.04.2016.A367) approved experiments.

The POI mouse model was induced by intestinal manipulation as described previously[6]. Animals were sacrificed 3 and 24 h after manipulation.

**Murine enteric glia cell cultures**. Primary enteric glial cell (EGC) cultures were obtained by sacrificing C57BL/6 or GFAPcre+ Ai14fl/fl mice, 8–16 weeks of age, extracting the small intestine, and cleansing it with 20 ml of oxygenated Krebs-

**Fig. 6 Enteric gliosis and IL-1-signaling are involved in acute intestinal inflammation after abdominal surgery. a** Immunohistochemistry for macrophages (CD68[+], green) and EGCs (GFAP[+], violet) in jejunal cross-sections. Hoechst (white) was used as counterstain. White arrows indicate ganglia-associated macrophages. Scale bar 50 µm. **b** Schematic overview showing the patient specimen collection of jejunal *ME* at an early and late time point during pancreatic head resection. **c** qPCR analysis of jejunal *ME* specimens for inflammation- and macrophage function-associated genes. $n = 10–15$; IL1β (14), IL-6 (15), CCL2 (13), CSF1 (10), CSF3 (11), CXCL2 (13). **d–g** Bulk-RNA-Seq analysis of jejunal *ME* specimens, including a PCA plot (**d**) and a Volcano plot visualizing differentially expressed genes (**e**) between late and early specimens ($p = 0.05$, fold: 1.5). $n = 4$. **f** GO term analysis of all differentially expressed genes between early and late specimens showed enrichment in gene clusters related to "glial activation", "IL-1-signaling", "migration", and "inflammatory response". **g** Heat map of significantly regulated genes highlighting gliosis-related genes. **h** Schematic overview of the generation and treatment of primary human EGCs from patient specimens. **i** *NanoString* analysis of primary human EGCs treated with IL1β or vehicle for 24 h. $n = 10$. Data are shown as mean RNA counts. Statistics were done with Student's *t*-test and Fisher's exact test. *<0.05, **<0.01, ***<0.001, All * compared to early *ME* specimens or untreated hEGCs.

Henseleit buffer (126 mM NaCl; 2.5 mM KCl: 25 mM NaHCO3; 1.2 mM NaH2PO4; 1.2 mM MgCl2; 2.5 mM CaCl2, 100 IU/ml Pen, 100 IU/ml Strep and 2.5 µg/ml Amphotericin). The small bowel was cut into 3–5 cm long segments and kept in oxygenated ice-cold Krebs-Henseleit buffer. Each segment was then drawn onto a sterile glass pipette, and the *ME* was stripped with forceps to collect muscle tissue for further digestion steps. After centrifugation (300xg for 5 min), the tissue was incubated for 15 min in 5 ml DMEM containing Protease Type1 (0.25 mg/ml, Sigma-Aldrich) and Collagenase A (1 mg/ml, Sigma-Aldrich) in a water bath at 37 °C, 150 rpm. The enzymatic digestion was stopped by adding 5 ml DMEM containing 10% FBS (Sigma-Aldrich), centrifugation for 5 min at 300xg, and re-suspended in proliferation medium (neurobasal medium with 100 IU/ Pen, 100 µg/ml Strep, 2.5 µg/ml Amphotericin (all Thermo Scientific), FGF and EGF (both 20 ng/ml, Immuno-tools). Cells in proliferation media were kept at 37 °C, 5% CO₂ for 4 days to promote the formation of enteric neurospheres. For experiments, enteric neurospheres were dissociated with trypsin (0.25%, Thermo Scientific) for 5 min at 37 °C and distributed at 50% confluency on Poly-Ornithine (Sigma-Aldrich) coated 6 well plates in differentiation medium (neurobasal medium with 100 IU/ Pen, 100 µg/ml Strep, 2.5 µg/ml Amphotericin, B27, N2 (all Thermo Scientific) and EGF (2 ng/ml, Immunotools). After 7 days in differentiation medium, mature enteric glia cells were treated with ATP (100 µM, Sigma) and IL-1 (10 ng/ml, Immunotools) and further processed for RNA isolation or their conditioned medium used for enzyme-linked immunosorbent assay (ELISA) or qPCR analysis.

**Murine bone-marrow-derived macrophage cultures.** Primary bone-marrow-derived macrophage (BMDM) cultures were obtained by sacrificing IL1R1[−/−] mice, 8–16 weeks of age, extracting the femoral bones and isolating bone-marrow stem cells with a syringe, and culturing the cells in RPMI supplemented with FCS (10%, Thermo Scientific), β-mercaptoethanol (1 mM, SIGMA) and 10 ng/ml rmCSF-1 (Immunotools). After 4 days in culture, the medium was changed to remove all dead cells, and on day 6, cells were used for all planned experiments.

For the scratch assays, BMDMs were transferred to 24-well plates and grown to 90% confluency. The "scratch" was performed with a 200 µl pipette tip, and damaged/dead cells were immediately removed by one wash step with PBS. BMDMs were treated with CM^Veh, CM^IL−1, neurobasal medium (NB) supplemented with 10% FCS, and NB without serum (control) for 24 h. After treatment, BMDMs were fixed with 4% PFA for 15 min and stained for F-Actin with Phalloidin FITC (Thermo Scientific) for 1 h, and processed with the fluorescence microscope Nikon TE2000 in the cell culture well plates.

For the transwell assays, BMDMs were placed in transwells (10,000 cells, Ibidi) and treated with CM^Veh, CM^IL−1, NB supplemented with 10% FBS, and NB without serum (control) for 3 h. Afterwards, transwells were fixed with 4% PFA for 15 min, and cells were stained for F-Actin with Phalloidin FITC (Thermo Scientific) and Hoechst for 1 h. Transwells were mounted on glass slights and processed with the fluorescence microscope Nikon TE2000.

For the phagocytosis assays, BMDMs were transferred to 24-well plates (10,000 cells/well) and treated with CM^Veh, CM^IL-1, neurobasal medium (NB) supplemented with 10% FBS and NB without serum (control) for 24 h. After removing the treatment medium, BMDMs were incubated with Dextran-Cascade Blue 10,000 MW (50 µg/ml, Thermo Scientific) for 1 h. Then washed two times with warmed PBS and collected by EDTA/Trypsin (0.05%) and processed by FACS analysis for high and low phagocytic cells.

**Human surgical specimens.** The ethics committee of the College of Medicine at the Ohio State approved the human IRB protocol[13] (Table S1).

The ethics committee of the University of Bonn, Germany, approved the collection of patient surgical specimens (*266_14*) (Table S2).

The human IRB protocol was approved by the ethics committee of the College of Medicine at The Ohio State University. Informed consent was obtained to procure viable human surgical tissue from the colon or small bowel from patients with polyps undergoing a colectomy (sigmoid colon) or patients undergoing Roux-en-Y by-pass surgery (jejunum) (Table S1). Human EGCs (hEGCs) in culture from 9 GI surgical specimens were used to study gene expression and IL1R1 immunoreactivity.

Human surgical tissue for the IOC experiments was collected from four patients undergoing pancreatectomy. Human jejunum specimens were collected in ice-cold oxygenated Krebs-Henseleit buffer during the surgical procedure and transported to the laboratory. Full-thickness jejunum specimens were incubated for 24 h with or without Anakinra (100 µg/ml) in DMEM/F12 with 10% FBS at 5%CO₂ and 37 °C. After 24 h, media were collected, centrifuged, and frozen in liquid nitrogen for ELISA analyses.

**Human EGC cultures.** Myenteric plexus tissue of patients was processed and cultured as described before[13,20]. Briefly, tissue collection was performed by the surgeon and immersed immediately in ice-cold oxygenated Krebs-Henseleit solution and promptly transported to the research facilities within 15 min in coordination with the Clinical Pathology Team. For isolating myenteric ganglia, tissue was pinned luminal side facing upwards under a stereoscopic microscope, and the mucosa, submucosa, and most of the circular muscle were dissected away using scissors and then flipped over to remove longitudinal muscle by dissection.

Myenteric plexus tissue was cut and enzymatically dissociated as described elsewhere (LIT) with modifications as follows: Myenteric plexus tissue was minced into 0.1–0.2 cm² pieces and dissociated in an enzyme solution (0.125 mg/ml Liberase, 0.5 µg/ml Amphotericin B) prepared in Dulbecco's modified Eagle's medium (DMEM)-F12, for 60 min at 37 °C with agitation. Ganglia were removed from the enzymatic solution by spinning down (twice), and re-suspending in a mixture of DMEM-F12, bovine serum albumin 0.1%, and DNase 50 µg/ml DNase (once). Solution containing the ganglia was transferred to a 100 mm culture dish, and isolated single ganglia free of smooth muscle or other tissue components were collected with a micropipette while visualized under a stereoscopic microscope and plated into wells of a 24-well culture plate and kept in DMEM-F12 (1:1) medium containing 10% fetal bovine serum (FBS) and a mixture of antibiotics (penicillin 100 U/ml, streptomycin 100 µg/ml, and amphotericin B 0.25 µg/ml) at 37 °C in an atmosphere of 5% CO₂ and 95% humidity.

After cells reach semi-confluence after 3–4 weeks (P1), hEGCs were enriched and purified by eliminating/separating fibroblasts, smooth muscle, and other cells. EGC enrichment and purification were achieved by labeling the isolated cells with magnetic microbeads linked to the anti-specific antigen, D7-Fib, and passing them through a magnetic bead separation column following the manufacturer's instructions (*Miltenyi Biotec Inc*, San Diego, CA). This purification protocol was performed twice (P2 and P3) to reach a cell enrichment of up to 10,000 fold, and 20,000 cells were plated on glass coverslips pre-coated with 20 µg/ml laminin/P-D-Lys in 50 mm bottom glass #0 culture dishes for immunostaining and imaging or 12-well plates for IL-6 or CCl2 release experiments. Cultured hEGCs were kept until confluent and harvested for additional experiments (4 to 10 days). On the day of the experiment, hEGCs were stimulated as indicated. Parallel to this, cells at each passage were split and seeded in plastic 25 mm² culture flasks and used for study in passages three to six.

**NanoString nCounter gene expression assay.** The RNA quality has been evaluated using Agilent RNA 6000 Nano Chip. NanoString nCounter technology is based on the direct detection of target molecules using color-coded molecular barcodes, providing a digital simultaneous quantification of the number of target molecules. Total (RNA 100 ng) was hybridized overnight with nCounter Reporter (20 µl) probes in hybridization buffer and excess of nCounter Capture probes (5 µl) at 65 °C for 16–20 h. The hybridization mixture containing target/probe complexes was allowed to bind to magnetic beads containing complementary sequences on the capture probe. After each target found a probe pair, excess probes were washed, followed by a sequential binding to sequences on the reporter probe. Biotinylated capture probe-bound samples were immobilized and recovered on a streptavidin-coated cartridge. The abundance of specific target molecules was then quantified using the nCounter digital analyzer. Individual fluorescent barcodes and target molecules in each sample were recorded with a CCD camera by performing a high-density scan (600 fields of view). Images were processed internally into a digital format and were normalized using the NanoString nSolver software analysis tool.

Counts were normalized for all target RNAs in all samples based on the positive control RNA to account for differences in hybridization efficiency and post-hybridization processing, including purification and immobilization of complexes. The average was normalized by background counts for each sample obtained from the average of the eight negative control counts. Subsequently, a normalization of mRNA content was performed based on internal reference housekeeping genes Gusb, TBP, NMNAT1, RBP1, STX1A, CTNNB1 using nSolver Software (Nano-String Technologies, Seattle, WA).

**Immunohistochemistry.** Whole-mount specimens were mechanically prepared by dissection of the (sub)mucosa, fixed in 4% paraformaldehyde/PBS for 30 min, permeabilized with 1% Triton-X 100/PBS for 15 min, blocked with 5% donkey serum/PBS for 1 h, and incubated with primary IgGs mentioned in appendix Table S4 at 4 °C overnight. After three PBS washing steps, secondary antibodies (Dianova, anti-rat IgG-Cy2 1:800, anti-guinea pig IgG-Cy3, anti-chicken IgY-FITC, and anti-rabbit IgG-FITC or - Cy3 1:800 were incubated for 90 min (Table S4). Specimens were mounted in Epredia Shando Immu-Mount (Thermo Scientific) and imaged on a Leica confocal imaging system or a Nikon 2000TE fluorescent microscope.

Primary cells were fixed in 4% paraformaldehyde/PBS for 30 min, permeabilized with 0.25% Triton-X 100/PBS for 15 min, blocked with 5% donkey serum in PBS for 1 h, and incubated with primary IgGs mentioned in Table S4 at 4 °C overnight.

After three PBS washing steps, secondary antibodies (Dianova, anti-mouse IgG-Cy2 1:800, anti-guinea pig IgG-FITC, and anti-rabbit IgG-FITC or - Cy3 1:800 were incubated for 60 min. Specimens were mounted in Fluorogel-Tris and imaged using a Leica confocal imaging system or a Nikon TE 2000 fluorescent microscope.

**Quantitative PCR.** Total RNA was extracted from *ME* specimens at indicated time points after IM using the RNeasy Mini Kit (Qiagen, Hilden, Germany), followed by deoxyribonuclease I treatment (Ambion, Austin, TX). Complementary DNA was synthesized using the High Capacity cDNA Reverse Transcription Kit (Applied Biosystems, Darmstadt, Germany). The expression of mRNA was quantified by real-time RT-PCR with primers shown in Table S3. Quantitative polymerase chain reaction was performed with SYBR Green PCR Master Mix (Applied Biosystems, Darmstadt, Germany).

**Flow cytometry (FACS).** FACS analysis was performed on isolated *ME* samples of the small bowel 24 h after IM in GFAP$^{Cre}$xIL1R1$^{fl/fl}$ animals. Isolation of *ME* was achieved by sliding small bowel segments onto a glass rod, removing the outer muscularis circumferentially with moist cotton applicators and cutting the *ME* into fine pieces. *ME* was digested with a 0.1% collagenase type II (Worthington Biochemical, Lakewood, NJ, USA) enzyme mixture, diluted in PBS, containing 0.1 mg/ml DNase I (La Roche, Germany), 2.4 mg/ml Dispase II (La Roche, Germany), 1 mg/ml BSA (Applichem), and 0.7 mg/ml trypsin inhibitor (Applichem) for 40 min in a 37 °C shaking water bath. Afterwards single-cell suspension was obtained using a 70 μm filter mesh and cells were stained for 30 min at 4 °C with the appropriate antibodies. For antibodies used in this study see Table S4. Flow cytometry analyses were performed on *FACSCantoI*(BD Biosciences) using *FACSDiva* software and data were analyzed with the latest *FlowJo* software (Tree Star, Ashland, OR, USA).

**Enzyme-linked immunosorbent assay (ELISA).** Release of IL-6 and CCL2 was measured in *ME* RIPA lysates isolated from small intestine segments at the indicated time points after IM. Release of IL-6 in EGC cultures incubated with various treatments was measured at the indicated time points. All ELISAs were purchased from (Thermo Scientific) and used according to the manufacturer's instructions. Values were normalized to tissue weights or untreated EGCs. Briefly, for animal tissue, the isolated *ME* (~50 mg) was lysed with 1xRIPA buffer for 30 min, centrifuged for 30 min at maximum speed and the protein concentration was determined with a BCA kit (Thermo Scientific). 100 μg of total protein was used to measure the release of IL-6 or CCL2 in duplicates. For EGCs, cells were treated with the indicated substances for 24 h, the supernatant was collected, centrifuged at 5000 rcf for 5 min, and snap-frozen in liquid nitrogen before being processed for the IL-6 or CCL2 ELISA.

**In vivo gastrointestinal transit.** Gastrointestinal transit (GIT) was assessed by measuring the intestinal distribution of orally administered fluorescently labeled dextran-gavage 90 min after administration as described previously[6]. The gastrointestinal tract was divided into 15 segments (stomach to the colon). The geometric center (GC) of labeled dextran distribution was calculated as described previously. The stomach (st) correlates with a GC of 1, the small bowel correlates with a GC of 2–11, the cecum (c) correlates with a GC of 12, and the colon correlates with a GC of 13–15. GIT measurements were performed with sham and IM24h animals of GFAP$^{Cre}$xIL1R1$^{fl/fl}$ or Sox10$^{CreERT2}$xRpl22$^{HA/+}$ animals.

**MPO$^+$-cell infiltration.** Jejunal mucosa-free ME whole-mount were fixed in ethanol and stained with Hanker Yates reagent (Polyscience Europe,

Germany) to identify myeloperoxidase expressing cells (MPO$^+$). The mean number of MPO$^+$ cells/mm$^2$ for 5 random areas per animal was determined. MPO$^+$ measurement was performed with animals 24h after IM.

**RNA-Seq.** RNA samples were extracted using the RNeasy Mini Kit (*Qiagen*). RNA-Seq libraries were prepared using the QuantSeq 3′ mRNA-Seq Library Prep Kit (*Lexogen*) according to the manufacturer's instructions by the Genomics Core Facility of the University Hospital Bonn. The method has high strand specificity (>99.9%), and most sequences are generated from the last exon and the 3′ untranslated region. The technique generates only one fragment per transcript, and the number of reads mapped to a given gene is proportional to its expression. Fewer reads than in classical RNA-seq methods are needed to determine unambiguous gene expression levels, allowing a high level of multiplexing. Library preparation involved reverse transcription of RNA with oligodT primers, followed by removal of RNA and second-strand cDNA synthesis with random primers. The resulting fragments containing both linker fragments were PCR amplified with primers containing the Illumina adaptors and sample-specific barcodes. All libraries were sequenced (single-end 50 bp) on one lane of the Illumina Hiseq 2500. Only genes with an adjusted *p*-value below 0.05 and a minimum fold-change greater than 1.5 were considered to be differentially expressed between conditions.

**RiboTag method.** *RiboTag* IP was performed according to Leven et al.[34]. Briefly, the muscle layer of the whole small bowel tissue was mechanically separated from the mucosal layer and placed in RNA*later* (Thermo Scientific). Muscle tissue was lysed on a Precellys homogenizer [Bertin Instruments] (3 × 5000 rpm, 45 s; 5 min intermediate incubation on ice) in pre-cooled homogenization buffer (50 mM Tris/HCl, 100 mM KCl, 12 mM MgCl₂, 1% NP-40, 1 mg/ml Heparin, 100 μg/ml Cycloheximide, 1 mM DTT, 200 U/ml RNAsin, 1× Protease Inhibitor P8340), centrifuged (10 min, 10,000 × g, 4 °C), and supernatants saved. "Input" controls were generated from 50 μl cleared lysate. Supernatants were incubated with anti-HA antibody (5 μl; 1 mg/ml; Table S4; 4 h, 4 °C, 7 rpm) and conjugates added to 200 μl of equilibrated A/G dynabeads (Thermo Scientific) and incubated (overnight, 4 °C, 7 rpm). High salt buffer (50 mM Tris/HCl, 300 mM KCl, 12 mM MgCl₂, 1% NP-40, 100 μg/ml Cycloheximide, 0.5 mM DTT) was used to wash beads before elution of cell-specific mRNA and subsequent mRNA extraction (Qiagen RNeasy micro kit).

**Software.** The software tools used for this study include Partek Flow, available from https://www.partek.com/partek-flow/#features; Subread/Feature Counts[61], available from http://subread.sourceforge.net/; Venn-Diagram Software, available from http://bioinformatics.psb.ugent.be/webtools/Venn/; and Gene Set Enrichment Analysis, available from https://www.partek.com/partek-flow/#features.

**Statistics and reproducibility.** Statistical analysis was performed with Prism V9.01 (GraphPad, USA) using Student *t*-test or one-way ANOVA as indicated. In all figures, *p*-values are indicated as *$p < 0.05$, **$p < 0.01$ and ***$p < 0.001$ when compared to control or #$p < 0.05$, ##$p < 0.01$ and ###$p < 0.001$ compared to indicated samples. All plots show the means of indicated expression levels ± standard error of the mean (SEM).

For all shown RNA-seq data, the Partek software was used for all analyses. Partek software performs statistical analyses by the Fisher's exact test and provides *p*-values with multiple testing corrections (FDR).

Experiments were repeated with more samples when the result was close to statistical significance, and sample sizes for animal studies were chosen following previously reported studies that have used the POI animal model; at least 6–10 independent mice per experimental setup. All animals were handled by standardized housing procedures and kept in precisely the same environmental conditions and were genotyped at 6 weeks of age and received a randomized number by which they were identified. Age- and sex-matched animals were grouped randomly and used in the POI animal model.

All the control or experimental mice in each experimental set were treated with the same procedure and manipulation. By this, we avoided any group or genotype-specific effects due to the timing of experiments or handling of animals.

**Reporting summary.** Further information on research design is available in the Nature Research Reporting Summary linked to this article.

## Data availability
The data sets produced in this study are available upon reasonable request and in the following databases: RNA-Seq data from mRNA of *ME*-tissue from patients in GSE149181. The *RiboTag* data of control and IM3h mice is available under the accession number *GSE198889*, and the data of EGCs treated with IL1β, and BMDMs treated with conditioned media is available under the accession number *GSE205610*. All raw data used for the main figures were included in an excel sheet named supplementary data 1.

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

## Acknowledgements

The authors thank the Next Generation Sequencing Core Facility and the Institute for Genomic Statistics and Bioinformatics of the University Clinics Bonn for supporting the RNA-Seq analysis. In addition, the authors thank the Flow Cytometry Core Facility of the University Clinics Bonn for supporting all FACS experiments. We thank the technicians Patrik Efferz, Mariola Lysson, Jana Müller, and Bianca Schneiker for their support with the readouts like ELISA and qPCR and for handling the transgenic mouse lines. We thank Prof. Vachilis Pachnis for sharing the *Sox10^{CreERT2}* mice with us. We thank PD Valentin Schäfer for sharing the Anakinra substance for in vitro assays. We thank Prof. Vanda A. Lennon for sharing the ANNA-1 antibody to visualize enteric neurons in our primary cell cultures. We thank Prof. Nico Schlegel and Prof. Wouter de Jonge for reading our manuscript and providing helpful suggestions for wording our results and conclusions. Graphical visualizations were created with BioRender software. We thank the following funding organizations for supporting our research: National Institutes of Health Grant (NCI Cost shared resource for COM, The Ohio State University): P30CA16058 National Institutes of Health, National Institutes of Diabetes, Digestive and Kidney Diseases (NIH, NIDDK) Grants: R01DK113943 and R01DK125809 (F.L.C.). BonnNI medical student Grant: Q-611.0754 (S.W.). ImmunoSensation2 Cluster of Excellence: EXC 2151–390873048 (S.W.). German research council (D.F.G.): WE4204/3-1 (S.W.).

## Author contributions

Conceptualization: R.S., F.L.C., and S.W. Methodology: R.S., J.C.K., F.L.C., P.L., S.M., M.B., T.G., Ph.L., T.O.V., and S.W. Investigation: R.S., P.L., M.B., S.M., L.S., E.Z., and P.F. Visualization: R.S., P.L., M.B., S.M., and P.F. Funding acquisition: J.C.K., F.L.C., and S.W. Project administration: J.C.K., F.L.C., and S.W. Patient material and clinical organization: T.G., Ph.L., T.O.V. Supervision: R.S., F.L.C., and S.W. Writing—original draft: R.S., J.C.K., F.L.C., and S.W. Writing—review & editing: R.S., J.C.K., F.L.C., P.L., S.M., T.G., Ph.L., T.O.V., L.S., M.B., and S.W.

## Funding

## Competing interests

The authors declare the following competing interests: S.W. and J.C.K. receive royalties from Wolter Kluwer for contributing to the postoperative ileus section of the *UpToDate* library. All other authors declare no competing interests.
