## [Peer Review File · Communications Biology]

Reviewers' comments:

Reviewer #1 (Remarks to the Author):

This manuscript reports on the pathophysiological interplay between IL-1R1 signalling, muscularis externa macrophages and enteric glial cells that leads to postoperative ileus (POI) after intestinal manipulation.

Using a wide range of different approaches (lab animals, organoids, human tissues) and techniques, the authors provide solid evidence for a IL-1 mediated macrophage-EGC interaction in the development of POI.

Reading the manuscript resulted in the following comments that may improve the manuscript:

General comment

IL-1 is likely not the sole cytokine involved in the immune cascade that leads to postoperative ileus. Therefore, the discussion would benefit from a brief comment on the specificity/relevancy of the role of IL-1 in respect to other important pro-inflammatory cytokines such as IL-6 or TNF α . This could include the applicability of the current results for the clinician in the surgery room: is there evidence that a preventive treatment with a IL-1 receptor antagonist during surgery may reduce abdominal surgery-induced postoperative ileus? Did the authors also test such a pharmacological treatment option (eg intra-abdominal application of an IL-1 inhibitor or anti IL-1 antibody) on intestinal manipulation-induced POI in mice?

Detailed comments:

1. Fig 4e: it is interesting that the inflammatory responses were less pronounced in the intestinal organotype cultures (IOC) from the muscularis externa from GFAP Cre+ xIL1R1fl/fl mice. But it is remarkable that a substantial response was still preserved in these IOC (see Figure 4e). This is somehow unexpected given the massive protective effect that was seen of IL-1R deficiency on macrophage activation and postoperative motility disturbances (Fig 3e). How do you explain this discrepancy. One expects that the effect in the relatively 'clean' organoids, which are devoid of blood-derived leukocytes, is comparable or even more pronounced than what you observe in muscularis externa tissue of IL-1R deficient mice.

2. The authors show that postoperative intestinal disturbances were almost absent in GFAP Cre+ xIL1R1fl/fl mice. Given the use of transgenic mice here, it would be interesting to know whether the gastrointestinal transit in the SHAM GFAP Cre- xIL1R1fl/fl mice was comparable to the GI transit that is seen in the non-transgenic C57BL/6 mice.

3. Line 334, human abdominal surgery: please specify what 'early' and 'late' means here, in terms of period of time that passed between the first (early) and second (late) tissue resection.

Reviewer #2 (Remarks to the Author):

This is an important and well written study demonstrating that IL-1 receptor signaling during enteric gliosis could play a central pathogenic role in driving paralytic ileus, which is an adverse and common side-effect of abdominal surgery. The investigators build on their earlier work which demonstrated that IL-1 R signaling is a key modulator of postoperative ileus (POI), by demonstrating that enteric glial cells (EGC) represent an upstream cellular target of proinflammatory cross-talk with muscularis externa macrophages that leads to dysmotility. Primary murine and human EGC/LLMP cultures exposed to mechanical trauma and/or IL1 triggers enteric gliosis that is recapitulated in vivo using a POI murine model, and early disease sequelae are linked to EGC using elegant genetic/ribosomal tagged and genetic ablation models to profile gene regulation and clinical outcomes associated with

disruption of IL-1R signaling in EGC. Linking disease sequelae to EGC-macrophage cross-talk is then performed using an in vitro model of POI, and clinical translation is indicated in resected jejunum from patients with pancreatic disease undergoing surgical procedures. This work potentially advances the field by conceptually identifying early cellular pathogenic cross-talk modulated by IL1R signaling as a candidate therapeutic target in POI. EGC are demonstrated as upstream regulators using specific genetic tools, which addresses prior weaknesses in the field and likely will be of broad interest to the journal readership. The authors should address the following items.

1. The concept of enteric gliosis and EGC dysfunction being a primary trigger of paralytic ileus is not new and the authors need to cite the two original reports [Bush et al. Fulminant jejuno-ileitis following ablation of enteric glia in adult transgenic mice. *Cell* 1998 93:189-201. Cornet A et al. Enterocolitis induced by autoimmune targeting of enteric glial cells: A possible mechanism in Crohn's disease. *Proc Natl Acad Sci USA* 2001 98:13306-13311]. These citations are especially important to include as they used similar genetic tools i.e. the gfap promoter to specifically target enteric glia, especially as both models demonstrated pathological sequelae that resulted in paralytic ileus using different mechanisms and genetic ablation strategies. The authors need to consider this work in the context of their own data since one study cited (#12 Rao et al *Gastro* 153, 2017) has questioned the specificity of gfap-targeted gene expression/ablation in the intestine. The authors need to justify the use of gfap-as an EGC specific promoter, since other cells types e.g. epithelial cells and lymphocytes could contribute to many of the signaling pathway proposed. Notably, the Rao study did not demonstrate pathology or ileitis, which has also been reported by the Sofoniew group in different environmental settings, likely due to microbiome compositional differences. In this regard, the present work is important since it implicates IL1R signaling as important triggers of EGC-induced pathology and could explain reported discrepancies in the field.
2. A potentially important limitation of the study is the uncertainty of knowing whether the ileus described is actually associated with pathological consequence, and more importantly whether the insult is merely delayed by ablating IL1R signaling in EGC. Further clarity in this regard seems warranted.
3. Demonstrating cellular specificity is also key in supporting the in vitro signaling interpretations using the gfap-promoter. Notably, the authors need to characterize potential cellular contaminants in the primary EGC cultures, since several other cells types may contribute to macrophage activation by conditioned media. The RNAseq data could provide useful information in this regard, especially since it is well known that EGC can transdifferentiate in culture.
4. The concept that LLMP dissection of mucosa and submucosa represents a novel POI model should be discussed in a more balanced manner. There is a plethora of studies working on smooth muscle using similar dissections that do not regard these dissections proinflammatory insult.
5. Supplemental methods and Tables were not provided so it was not possible to evaluate the methodology and statistical approaches. Insufficient detail of patient resected tissues were provided, especially medical interventions that could have impacted the study findings. It is not clear whether multiple testing corrections were applied to support the RNAseq findings.
6. Key gene annotations should be added to several figures e.g. Fig 1f

Reviewer #3 (Remarks to the Author):

The manuscript by Schneider et al investigates the unique features of IL-1-induced enteric gliosis and its role in modulating macrophages of relevance for development of surgery induced inflammation and dysmotility. It is first demonstrated that the RNA profile of IL-1b-induced enteric gliosis resembles the profile induced by intestinal manipulation (modelling surgery-induced inflammation) and includes genes involved in immune cell migration and inflammation. It is then demonstrated that IL1-dependent immune responses (including macrophage activation and ganglionic infiltration) as well as motility defects depends on IL1R1 in enteric glia. These results are also demonstrated in isolated guts (that cannot be infiltrated by blood-borne immune cells) indicating the importance of resident macrophages in the process. Further evidence that IL1 creates a unique glial response and

downstream effects on macrophages are provided by treatment of cultivated macrophages with IL1b-stimulated enteric glia conditioned media. Last, a similar response (involving genes mediating inflammatory response, IL-1 signaling and glial activation) is identified in intestine sampled from patients undergoing abdominal surgery. The main conclusions are that IL-1-induced glia trigger dysmotility and an inflammatory response involving macrophages in mice, which probably is conserved in the human.

Overall, this is a comprehensive and meticulously carried out study that additionally is very well-presented. Previous work is accredited and incorporated into an interesting discussion, which also provide several directions for future investigations. Nevertheless, some issues (summarized below) need to be addressed to lend full credibility to some of the claims made, including the causative role of macrophages on gut dysmotility.

Major concerns:

- 1) Genes identified after IL1b-induction are considered unique, but this is only compared towards previous ATP-induced gliosis. It would be helpful to understand how it compares and possibly differs also from LPS or IFN γ treatment (as described in for instance Chow et al., CMGH 2021). For instance, CSF is reported upregulated as response to LPS.
- 2) 20% gene overlap between POI and IL1b induced gliosis is reported -please include the list of genes as a table.
- 3) How have you validated that your "enteric glial cultures" generated from wildtype mice or human tissue mainly contains glia and not neurons? At least murine enteric neurons express Il1r1 (IPANs, see mousebrain.org) and could therefore also contribute to the "reactive glial" profile in the experiments described in 6h. Given other supportive data, especially where enteric glia specific deletion of Il1r1 is included, your data is rather convincing, but formally there could be neuronal contribution to the results presented especially in 6h which should be discussed. Moreover, if neurons are present in cultures presented in Figure 5a, you could partially be looking at indirect signalling through neurons that respond to the enteric glia (the response of which could depend on whether glia have responded to Il1r1 or not).
- 4) Is there any evidence provided that the resolved GI-transit phenotype depends on the mild enteric macrophage induction in GFAP-Cre IL1R1 $^{fl/fl}$ animals? An alternative explanation could be that peristalsis-controlling enteric neurons are directly influenced by IL1R1-induced enteric glia. If there is no causative link between macrophages and neurons provided in your or previous studies that you can refer to, the title should be amended to reflect this possibility (it implies that dysmotility is caused by a modulated macrophage function, while it is possible that these are parallel mechanisms).
- 5) Previous studies have shown that MHC-II is induced in enteric glia when they enter an active state (as also mentioned in your discussion). It would be interesting if you could report whether MHC-II is induced also in your IL1B induced and/or POI-induced enteric gliosis.
- 6) Overall, this manuscript includes many heatmaps (Fig S1f, Fig 2f,g, S6d and more) only to demonstrate difference between control and treated states. Please include a subset of genes in each of those to make them more informative.

Minor concerns:

1) The coloured headings in Figure 2f,g, 3d, S6d (and more, go through all) are so dark that the text is not visible. Please lighten the colour. It would also be advisable to change the colour coding in the cases where control and treated are distinguished by the same colours as the z-score (red-blue), as this could be confusing.

2) Row 392. The sentence needs editing. "We and others have recently been shown to be suitable for generating an in vivo snapshot analysis of actively transcribed RNA selectively in EGCs". Although the study is overall well written and easy to follow, there are some more sentences throughout the manuscript that needs attention as well, please revise.

Responses to Reviewers

Dear Reviewers:

Thank you for your positive comments and recommendations to revise and improve our manuscript. All questions and concerns have been addressed, and the manuscript has been revised accordingly, including major clarifications, additional data, and extensive revision of the discussion. A point-by-point response is provided for each question raised by the reviewers.

Referee expertise:

Referee #1: Models of intestinal inflammation

Referee #2: Enteric nervous system and bacterial infection

Referee #3: Enteric nervous system

Reviewers' comments:

Reviewer #1 (Remarks to the Author):

This manuscript reports on the pathophysiological interplay between IL-1R1 signalling, muscularis externa macrophages and enteric glial cells that leads to postoperative ileus (POI) after intestinal manipulation.

Using a wide range of different approaches (lab animals, organoids, human tissues) and techniques, the authors provide solid evidence for a IL-1 mediated macrophage-EGC interaction in the development of POI.

Reading the manuscript resulted in the following comments that may improve the manuscript:

General comment

A) IL-1 is likely not the sole cytokine involved in the immune cascade that leads to postoperative ileus. Therefore, the discussion would benefit from a brief comment on the specificity/relevancy of the role of IL-1 in respect to other important pro-inflammatory cytokines such as IL-6 or TNF α .

- The reviewer is correct; IL-1 is only one of the many important pro-inflammatory cytokines in POI, so we included additional sentences of other relevant cytokines in the discussion. (see lines 424-432)

B) This could include the applicability of the current results for the clinician in the surgery room: is there evidence that a preventive treatment with a IL-1 receptor antagonist during surgery may reduce abdominal surgery-induced postoperative ileus?

- To our knowledge, there is no published or ongoing clinical trial with an IL1R-antagonist aiming to investigate any effects on postoperative motility disturbances. However, for intestinal diseases, there are multiple trials with Anakinra, a commercially available antagonist, also known as Kineret, to validate anti-inflammatory effects on, e.g., mucositis (NCT03233776) and colorectal cancer (NCT02090101). Availability of the drug Anakinra for clinical trials provides an opportunity to design a trial as a preventative treatment for postoperative ileus in abdominal surgery patients, based on the additional evidence provided in our current study to support IL1R1 signaling as a therapeutic target in postoperative ileus.

Due to this knowledge gap, we also decided to add another *in vitro* experiment to study whether Anakinra can block key mediator release by human ME, i.e., IL-6 and CCL2 after mechanical manipulation. We used human small bowel specimens, freshly collected from jejunal resection material of surgical patients, and treated them with Anakinra for 24 hours in an *in vitro* setting. We recently used this intestinal organoid culture (IOC) system in a similar way to prove EGC activity in human gut specimens (PMID: 33332729). After the Anakinra treatment, we measured the release of IL-6 and CCL2, which we have shown to be released by EGCs, to investigate the inflammatory state after the surgical manipulation. We included these new results in **figure S6e** and discussed them in the context of the potential preventative use of Anakinra in POI. (see lines 574-584)

C) Did the authors also test such a pharmacological treatment option (eg intra-abdominal application of an IL-1 inhibitor or anti IL-1 antibody) on intestinal manipulation-induced POI in mice?

- Our previous publications, Stoffels et al. (PMID: 24067878) and Hupa et al. (PMID: 24067878), proved the importance of IL-1-signaling. In Stoffels et al., a commercially available antagonist for IL1R1, Anakinra, had a positive impact on the clinical outcome (decreased GI transit time and improved motility) in our POI mouse model. Our present study links these beneficial effects to glial IL-1-signaling in POI development. Furthermore, by using human IOCs, we are now able to transfer our findings to the human intestine

in vitro and IL-1R1 antagonism could inhibit small bowel release of IL-6 and CCL2; both known to be released from reactive EGC after surgery. We further elaborated on these effects in the discussion part (in line with the discussion on comment B) to present a more comprehensive picture of the role of IL-1 and EGCs. (see lines 574-584)

Detailed comments:

1. Fig 4e: it is interesting that the inflammatory responses were less pronounced in the intestinal organotype cultures (IOC) from the muscularis externa from GFAP Cre+ xIL1R1fl/fl mice. But it is remarkable that a substantial response was still preserved in these IOC (see Figure 4e). This is somehow unexpected given the massive protective effect that was seen of IL-1R deficiency on macrophage activation and postoperative motility disturbances (Fig 3e). How do you explain this discrepancy. One expects that the effect in the relatively ‘clean’ organoids, which are devoid of blood-derived leukocytes, is comparable or even more pronounced than what you observe in muscularis externa tissue of IL-1R deficient mice.

See also Reviewer 2, Q4

- We thank the reviewer for this interesting comment. Although transcriptional responses measured at the early 3h time point might not per se allow a linear extrapolation on the functionality measured at the later 24h time point, we believe that there might be additional mechanistic insight by using the model of intestinal organotypic cultures (IOCs). Compared to the relatively gentle *in vivo* intestinal manipulation of the *muscularis externa* (ME), the ME specimens used for the IOCs were created by a more substantial mechanical stimulus, i.e., the mechanical separation procedure of the ME from the underlying mucosa is a much more severe surgical trauma than that inflicted on the animals during gut manipulation. Furthermore, although the ME remains intact, the blood vessels and the extracellular matrix connecting both layers are disrupted. Neural connections between myenteric and submucous plexus, as well as intrinsic primary afferent neurons projecting to the mucosa, are severed and disrupted. We hypothesized that this mechanical procedure leads to a stronger inflammatory response and performed qPCR measurements of several mediators to compare the innate immune response. Indeed, several cytokines, as well as macrophage and EGC activation markers, were more strongly induced in the IOCs than in the *in vivo* manipulated IM specimens after 3h (see below). In the revised manuscript, we discussed this observation and included the new data in **figure S4a**. (see lines 282-287)

It should be noted that we are confident that the IL-1 activation of EGCs is a crucial factor in POI pathogenesis. Still, we are also aware that other factors can induce EGC reactivity and thus mask the effects in the IOCs of conditional KO mice.

2. The authors show that postoperative intestinal disturbances were almost absent in GFAP Cre+ xIL1R1fl/fl mice. Given the use of transgenic mice here, it would be interesting to know whether the gastrointestinal transit in the SHAM GFAP Cre-xIL1R1fl/fl mice was comparable to the GI transit that is seen in the non-transgenic C57BL/6 mice.

- We performed sham surgeries in transgenic mice (GFAP^{Cre}- IL1R1^{fl/fl}) and C57BL/6 mice to compare the GI-transit (**Revision-figure A**, see at the end of this point by response). The data show no difference in motility between both groups. We mentioned this observation in the results part but did not include it in figure 3 as it is not a key finding.

3. Line 334, human abdominal surgery: please specify what ‘early’ and ‘late’ means here, in terms of period of time that passed between the first (early) and second (late) tissue resection.

- All needed information on the patient samples is documented in the supplementary file within table S2. In short, during the pancreaticoduodenectomy procedure, the “early” samples are collected after the first surgical transaction of the small intestine, while the “late” samples were taken from the same region (proximal end of the lower transected jejunum) immediately before reconstruction of small bowel passage after pancreas head resection. All samples were collected in ice-cold, oxygenated Krebs buffer and processed immediately. The time duration between the collections of the human specimens is also documented in table S2.

Reviewer #2 (Remarks to the Author):

This is an important and well written study demonstrating that IL-1 receptor signaling during enteric gliosis could play a central pathogenic role in driving paralytic ileus, which is an adverse and common side-effect of abdominal surgery. The investigators build on their earlier work which demonstrated that IL-1 R signaling is a key modulator of postoperative ileus (POI), by demonstrating that enteric glial cells (EGC) represent an upstream cellular target of proinflammatory cross-talk with muscularis externa macrophages that leads to dysmotility. Primary murine and human EGC/LLMP cultures exposed to mechanical trauma and/or IL1 triggers enteric gliosis that is recapitulated in vivo using a POI murine model, and early disease sequelae are linked to EGC using elegant genetic/ribosomal tagged and genetic ablation models to profile gene regulation and clinical outcomes associated with disruption of IL-1R signaling in EGC. Linking disease sequelae to EGC-macrophage cross-talk is then performed using an in vitro model of POI, and clinical translation is indicated in resected jejunum from patients with pancreatic disease undergoing surgical procedures. This work potentially advances the field by conceptually identifying early cellular pathogenic cross-talk modulated by IL1R signaling as a candidate therapeutic target in POI. EGC are demonstrated as upstream regulators using specific genetic tools, which addresses prior weaknesses in the field and likely will be of broad interest to the journal readership. The authors should address the following items.

1. The concept of enteric gliosis and EGC dysfunction being a primary trigger of paralytic ileus is not new and the authors need to cite the two original reports [Bush et al. Fulminant jejuno-ileitis following ablation of enteric glia in adult transgenic mice. *Cell* 1998 93:189-201. Cornet A et al. Enterocolitis induced by autoimmune targeting of enteric glial cells: A possible mechanism in Crohn's disease. *Proc Natl Acad Sci USA* 2001 98:13306-13311].

Thank you for pointing that out. We included the two publications in the introduction part and discussed them accordingly.

These citations are especially important to include as they used similar genetic tools i.e. the gfap promoter to specifically target enteric glia, especially as both models demonstrated pathological sequelae that resulted in paralytic ileus using different mechanisms and genetic ablation strategies. The authors need to consider this work in the context of their own data since one study cited (#12 Rao et al *Gastro* 153, 2017) has questioned the specificity of gfap-targeted gene expression/ablation in the intestine. The authors need to justify the use of gfap-as an EGC specific promoter, since other cells types e.g. epithelial cells and lymphocytes could contribute to many of the signaling pathway proposed.

Notably, the Rao study did not demonstrate pathology or ileitis, which has also been reported by the Sofoniew group in different environmental settings, likely due to microbiome compositional differences. In this regard, the present work is important

since it implicates IL1R signaling as important triggers of EGC-induced pathology and could explain reported discrepancies in the field.

- We agree with this reviewer that there are contradictory findings on the role of EGCs in the maintenance of barrier integrity and development of spontaneous bowel wall inflammation, as shown within the mentioned publications. Indeed Rao et al. showed by immunohistochemical and immunofluorescence staining in GFAP^{CreER} mice, 15 days after tamoxifen injection, the expression of GFAP in some rare epithelial cells, that were not further characterized. It should be noted that our study exclusively focused on the *muscularis externa* (ME), since postoperative inflammation is mostly restricted to this layer, while *lamina propria* and the mucosa are hardly affected (PMID: 22122661). Although we cannot exclude a role of rare GFAP-expressing epithelial cells, we strongly believe that the GFAP-driven IL1R1 knockout phenotype in POI is most likely based on IL1R1 deficiency of a major subpopulation of EGCs that express GFAP in the intestine. Notably, another study with a GFAP-promotor-driven transgenic mouse line also showed distinct effects of EGCs on intestinal motility by analyzing conditional Connexin43-KO mice (PMID: 24211490). (see lines 69-79)

We have also validated this in our GFAP-Cre mouse model by breeding GFAP^{Cre+} mice with a td-tomato (tdT) floxed reporter strain (Ai14^{fl/fl}) and quantified SOX10⁺/td⁺ cells in the small intestine and colon. All tdT⁺ cells within the ME are SOX10⁺, and only a minority of td-tomato cells are not labeled by SOX10 (see below), indicating that our conditional IL1R1-KO mouse is efficient in genetically deleting IL1R1-signaling in almost all GFAP⁺ EGCs. The few SOX10⁻ td-tomato⁺ cells could be neurons or ENS progenitor cells that are de-differentiated glia cells. However, this small number should not influence our results and the corresponding conclusions. This quantification is added to the manuscript in **figure S3b**. Furthermore, we noted contradictory findings on the role of different genetically driven EGC depletion models on barrier homeostasis in the introduction of the revised manuscript.

b

2. A potentially important limitation of the study is the uncertainty of knowing whether the ileus described is actually associated with pathological consequence, and more importantly whether the insult is merely delayed by ablating IL1R signaling in EGC. Further clarity in this regard seems warranted.
- This is indeed an important aspect. To answer this question, we performed additional animal experiments. We added a later time point, IM72h, to gain a more comprehensive view of the disease progression and, most importantly, the recovery stage. These new data were included in **figure 3e** and depicted a clear recovery of GI motility in both groups. The recovery in conditioned IL-1R1-KO mice proves that there is no delay in the POI development at later stages and that the deleted glial IL1R1 signaling protects mice from motility impairment in the inflammatory phase (24h) of this disease.

3. Demonstrating cellular specificity is also key in supporting the in vitro signaling interpretations using the gfap-promoter. Notably, the authors need to characterize potential cellular contaminants in the primary EGC cultures, since several other cells types may contribute to macrophage activation by conditioned media. The RNAseq data could provide useful information in this regard, especially since it is well known that EGC can transdifferentiate in culture.

See also Reviewer 3, Q3

- We are aware that our glial cell cultures are not 100% pure. This is true for many other studies dealing with primary material isolated from solid tissue by enzymatic digestions. Indeed, by counting SOX10⁺ cells, we showed a purity of more than 80% (EGCs/Total cells) with our established and published culture protocol (PMID: 33332729 + 31332247). Notably, we set this as the minimum acceptable threshold in all EGC experiments. Inspired by this comment, we added neuronal markers to our routinely used alpha-smooth muscle actin (α SMA) staining that identifies smooth muscle cells that are the majority of cell contaminants in our EGC cultures. Neuronal markers tested herein were MAP2, and ANNA1 and immunofluorescence stainings hardly detected any enteric neurons; only very few single neurons were found (**Figures S5a**).

Furthermore, we examined our RNA-Seq data of EGC cultures for the expression of glial, neuronal, and non-ENS cell markers. The latter included particular markers of macrophages, epithelial and endothelial cells. The RNA-

Seq data showed a clear enrichment in glial markers and only an extremely low (neuronal markers, the epithelial marker villin, and ICAM) and often even no detection of mRNAs of non-ENS cell markers (especially for macrophage markers F4/80, CD11b, CD11c, MHCII) (new **Figure S5b**). Therefore, from the transdifferential aspect, we can exclude a “contamination” and significant contribution of transdifferentiated neuronal cells and, importantly also of macrophages, while some inputs from the smooth muscle cell might still be possible. However, we would like to pronounce that, actually, all studies we are aware of either did not check the purity or also showed smooth muscle cell contamination within primary EGC cell cultures. We believe that our protocol and routine batch quality controls ensured the use of EGC cultures with high and acceptable purity to support the claims of this manuscript.

- The concept that LLMP dissection of mucosa and submucosa represents a novel POI model should be discussed in a more balanced manner. There is a plethora of studies working on smooth muscle using similar dissections that do not regard these dissections proinflammatory insult.

See also Reviewer 1, Q1

- The model of intestinal organotypic cultures (IOCs) of *the muscularis externa* (ME) is indeed not novel and often used in physiological and pharmacological studies with calcium wave or other neuronal activity measurements as a standard readout (PMID: 31956167). To our knowledge, the immune response of the isolated ME at an early time point (3 hours after incubation at 37°C/5% CO₂) has not been performed yet. Our data showed that the *ex vivo* mechanical manipulation (preparation of the tissue) induces inflammatory mediator gene expression, especially IL-1 β and activation markers for EGCs and macrophages. This induction even outranges the induction of our *in vivo* IM/POI model in mice (new **Figure S4a**). Therefore, researchers should consider that the mechanically induced local inflammatory response might impact *ex vivo* measurements, including the above-mentioned physiological and pharmacological studies. (see lines 282-287)

Notably, we do not believe that IOCs “represent “a novel *ex vivo* POI model”, as these cultures are missing some crucial characteristics of the animal model necessary for POI development. Major missing features are the missing peripheral innervation by the PNS and SNS and, as mentioned before, the connection to the blood circulation and the absence of any blood-derived immune cell infiltration. However, they represent a suitable model to study local tissue immune responses to trauma without blood-derived immune cell infiltration. As suggested by this reviewer, we discussed this topic in a more balanced manner in the revised manuscript. (see lines 476-493)

- Supplemental methods and Tables were not provided so it was not possible to evaluate the methodology and statistical approaches. Insufficient detail of patient resected tissues were provided, especially medical interventions that could have impacted the study findings. It is not clear whether multiple testing corrections were applied to support the RNAseq findings.

- We apologize that the reviewer had no access to the supplementary methods and tables, as we submitted all information in a supplementary file. Moreover, we included additional information for the applied statistics and methodology and provided more details about the patient samples (e.g., previous illnesses and medications) in table S2.

For our RNAseq data, we used the Partek software for all analyses. This software performs statistical analyses by the *Fisher's exact test* and provides p -values with multiple testing corrections (FDR) between, for example, the two groups: late and early intestinal patient specimens. We added this information to the methods part.

6. Key gene annotations should be added to several figures e.g. Fig 1f

- We added key genes to all heat maps of our study.

Reviewer #3 (Remarks to the Author):

The manuscript by Schneider et al investigates the unique features of IL-1-induced enteric gliosis and its role in modulating macrophages of relevance for development of surgery induced inflammation and dysmotility. It is first demonstrated that the RNA profile of IL-1b-induced enteric gliosis resembles the profile induced by intestinal manipulation (modelling surgery-induced inflammation) and includes genes involved in immune cell migration and inflammation. It is then demonstrated that IL1-dependent immune responses (including macrophage activation and ganglionic infiltration) as well as motility defects depends on IL1R1 in enteric glia. These results are also demonstrated in isolated guts (that cannot be infiltrated by blood-borne immune cells) indicating the importance of resident macrophages in the process. Further evidence that IL1 creates a unique glial response and downstream effects on macrophages are provided by treatment of cultivated macrophages with IL1b-stimulated enteric glia conditioned media. Last, a similar response (involving genes mediating inflammatory response, IL-1 signaling and glial activation) is identified in intestine sampled from patients undergoing abdominal surgery. The main conclusions are that IL-1-induced glia trigger dysmotility and an inflammatory response involving macrophages in mice, which probably is conserved in the human. Overall, this is a comprehensive and meticulously carried out study that additionally is very well-presented. Previous work is accredited and incorporated into an interesting discussion, which also provide several directions for future investigations. Nevertheless, some issues (summarized below) need to be addressed to lend full credibility to some of the claims made, including the causative role of macrophages on gut dysmotility.

Major concerns:

1) Genes identified after IL1b-induction are considered unique, but this is only compared towards previous ATP-induced gliosis. It would be helpful to understand how it compares and possibly differs also from LPS or IFN γ treatment (as described in for instance Chow et al., CMGH 2021). For instance, CSF is reported upregulated as response to LPS.

- We agree that it might be of particular interest to scientists to understand more about core signatures (overlapping genes) as well as unique patterns of EGC reactivity in multiple inflammatory conditions and diseases. By screening the literature and validating available data, we realized that this topic would alone deserve a separate manuscript to properly address this important question, and we feel it is beyond the scope of the current study. Therefore we didn't perform new experiments but to begin to address this question, we did compare published RNAseq datasets with our dataset in an attempt to describe the IL1R1-dependent gliosis signature in EGCs more precisely. We used data from a virus infection model in which IFN γ is also strongly induced as well

as a DNBS colitis model to activate EGCs. Unfortunately, Chow et al. did not provide supplementary data files with quantifications, so we could not include this data in our analyzes. Instead, we created a Venn diagram (see below, now incorporated as supplementary **figure S1g**) and compared the available data with our IL-1 β data. The list of the overlapping and unique genes included in our gliosis filter was added to the supplementary file as **table S5**. We rephrased and discussed this topic more extensively in the revised manuscript and avoided the term “unique “ as more comparative conditions are needed to identify real unique signatures. (see **lines 453-460**)

g

2) 20% gene overlap between POI and IL1b induced gliosis is reported -please include the list of genes as a table.

- We added this comparison to the supplementary file as **table S6**.

3) How have you validated that your “enteric glial cultures” generated from wildtype mice or human tissue mainly contains glia and not neurons? At least murine enteric neurons express Il1r1 (IPANs, see mousebrain.org) and could therefore also contribute to the “reactive glial” profile in the experiments described in 6h. Given other supportive data, especially where enteric glia specific deletion of Il1r1 is included, your data is rather convincing, but formally there could be neuronal contribution to the results presented especially in 6h which should be discussed. Moreover, if neurons are present in cultures presented in Figure 5a, you could partially be looking at indirect signalling through neurons that respond to the enteric glia (the response of which could depend on whether glia have responded to Il1r1 or not).

See also Reviewer 2, Q3

- We are aware that our glial cell cultures are not 100% pure. This is true for many other studies dealing with primary material isolated from solid tissue by enzymatic digestions. Indeed, by counting SOX10⁺ cells, we showed a purity of more than 80% (EGCs/Total cells) with our established and published culture protocol (PMID: 33332729 + 31332247). Notably, we set this as the minimum acceptable threshold in all EGC experiments. Inspired by this comment, we added neuronal markers to our routinely used alpha-smooth muscle actin (α SMA) staining protocols that identify smooth muscle cells that are the main (but minor) contaminant of our EGC culture. Neuronal markers tested herein were MAP2 and ANNA1, and immunofluorescence stainings hardly detected any enteric neurons; only very few single neurons were detected (**Figures S5a**).

Furthermore, we examined our RNA-Seq data of EGC cultures for the expression of glial, neuronal, and non-ENS cell markers. The latter included particular markers of macrophages, epithelial and endothelial cells. The RNA-Seq data showed a clear enrichment in glial markers and only an extremely low (neuronal markers, the epithelial marker villin and ICAM) and often there was no detection of mRNAs for non-ENS cell markers (especially for macrophage markers F4/80, CD11b, CD11c, MHCII) (new **Figure S5b**). Therefore, in terms of the transdifferentiation, we can exclude neuronal “contamination” and the significant contribution of transdifferentiated neuronal cells; equally as important, we can exclude macrophages as contaminants, while some smooth muscle cell contamination is still possible. However, we would like to emphasize that, actually, all studies we are aware of either did not check the purity at the molecular level as done herein or also had to accept a minor level of smooth muscle cell contamination that is present within primary EGC cell cultures. We believe that our protocol and routine batch quality controls ensured the use of EGC cultures with high and acceptable purity to support the claims of this manuscript.

For the human EGCs, we also performed ICC for neuronal markers showing only limited amounts of enteric neurons in the early culture passages and no neurons in the later EGC culture passages. We included these images as **revision-figure B** (see at the end of this point by response). It should be noted that isolation of single ganglia and harvesting of each ganglion under high magnification and visual inspection by an experienced researcher allows for the selection of nervous tissue (i.e., exclusively ganglia), excluding all other non-ganglion cell types for further purification using immune-isolation and double purification in P1 and P2 passages (as described in the manuscript/references). The only remaining contaminant is possibly neurons in the ganglia (1:7 ratio, neurons:glia) harvested]. However, the pan-neuronal marker HuC/D does not label any neurons in hEGC cultures used for IL1-signaling experiments.

Furthermore, to validate the EGC release of IL-1-induced factors, we treated EGC cultures from IL1R1^{fl/fl} x GFAP^{Cre+} and IL1R1^{fl/fl} x GFAP^{Cre-negative} mice

with IL-1. ELISA and qPCR measurements validated a diminished mRNA expression and protein release of the inflammatory factors (e.g., *Ccl2*, *IL-6*, *Ccl5*, *Cxcl5*, *Csf1*) established in our EGC cultures from conditional IL1R1-KO mice. This raises confidence that the purity of mEGC cultures is sufficiently high to study glial IL-1 induction of various inflammatory factors. We included this data in the supplementary **figure S5d+e** and added the findings to the results section. (see **lines 323-328**)

4) Is there any evidence provided that the resolved GI-transit phenotype depends on the mild macrophage induction in GFAP-Cre IL1R1^{fl/fl} animals? An alternative explanation could be that peristalsis-controlling enteric neurons are directly influenced by IL1R1-induced enteric glia. If there is no causative link between macrophages and neurons provided in your or previous studies that you can refer to, the title should be amended to reflect this possibility (it implies that dysmotility is caused by a modulated macrophage function, while it is possible

that these are parallel mechanisms).

- The reviewer is correct; we deduced the connection between milder macrophage activation and the impact on motility from our previous work. Accordingly, we changed our title from “IL-1-dependent enteric gliosis guides intestinal inflammation and dysmotility **by** modulating macrophage function” to “IL-1-dependent enteric gliosis guides intestinal inflammation and dysmotility **and modulates** macrophage function”.

Moreover, we included an additional section in our discussion to emphasize the direct impact of EGCs on enteric neuron function, e.g., motility and homeostasis. (see lines 537-547)

5) Previous studies have shown that MHC-II is induced in enteric glia when they enter an active state (as also mentioned in your discussion). It would be interesting if you could report whether MHC-II is induced also in your IL1B induced and/or POI-induced enteric gliosis.

- That is an interesting point; we tested IL-1 β -treated EGC cultures and whole specimens prepared from IM24h mice for the expression of MHC-II by immunohistochemistry. *In vitro*, we detected no MHC-II expression in EGC cultures, and *in vivo*, we could see no GFAP⁺ cells expressing MHC-II (see **Revision figure below**). Moreover, we could not find any expression of the MHC-II genes, H2-Ea and H2-Eb1, in our RNA-Seq dataset of EGC cultures with or without IL-1 β treatment and the POI-IM3h *RiboTag* mice.

6) Overall, this manuscript includes many heatmaps (Fig S1f, Fig 2f,g, S6d and more) only to demonstrate difference between control and treated states. Please include a subset of genes in each of those to make them more informative.

See also Reviewer 2, Q6

- We added key genes to all heat maps of our study.

Minor

concerns:

1) The coloured headings in Figure 2f,g, 3d, S6d (and more, go through all) are so dark that the text is not visible. Please lighten the colour. It would also be advisable to change the colour coding in the cases where control and treated are distinguished by the same colours as the z-score (red-blue), as this could be confusing.

- We appreciate the comment and modified the figures for better visibility and understanding.

2) Row 392. The sentence needs editing. “We and others have recently been shown to be suitable for generating an in vivo snapshot analysis of actively transcribed RNA selectively in EGCs”. Although the study is overall well written and easy to follow, there are some more sentences throughout the manuscript that needs attention as well, please revise.

- We read the manuscript more carefully, had a native speaker review our sentences, and amended complicated or incomprehensible sentences.

Figure Revision: **A)** GI-transit analysis of Bl6 mice and Cre-negative IL1R1-floxed mice after sham operation. GI-motility was similar in both groups after the sham operation. **B)** Immunocytochemistry of human EGC cultures for glia (S100 β , red) and neuron (ANNA1, green) markers. DAPI (blue) was used for nuclei staining. Only small numbers of enteric neurons (white arrows) were detected in the early passage of hEGC cultures. Later passages showed no enteric neurons. Scale bar = 100 μ m. **C)** Immunohistochemistry of small intestine ME of IM24h mice for glia (GFAP, red) and MHCII-expressing cells (green). Hoechst (blue) was used for nuclei staining. No MHC-II-expressing glia were detected in IM24h-ME samples. Scale bar = 100 μ m. Data are shown as mean \pm SEM. ***<0.001, *<0.05 * were compared to marked controls.

REVIEWERS' COMMENTS:

Reviewer #1 (Remarks to the Author):

Thank you for the answers and additional experimentations in response to the questions that were raised. I have no further comments.

Reviewer #2 (Remarks to the Author):

The authors have adequately addressed reviewer comments.

Reviewer #3 (Remarks to the Author):

The revised manuscript is substantially improved and all previous concerns have been resolved.